# QMoE+: Hybrid Quantum Mixture of Experts

## Abstract

Quantum mixture of experts (QMoE) extends conditional computation to the NISQ setting by distributing learning across parameterized quantum circuit (PQC) experts selected via a routing mechanism. Existing approaches are limited by single-block experts, lack of load balancing, and aggregation schemes that ignore routing amplitudes. We propose **QMoE+**, which uses two-block data re-uploading experts with learnable offsets, a coherent aggregation circuit over the joint routing-data Hilbert space, and a Switch-style load-balancing loss. Under top-$k$=1 sparse routing, QMoE+ activates only ∼28% of its parameters per inference while achieving consistent accuracy gains across seven datasets and four gate sets, winning 27 of 28 configurations (7 datasets × 4 gate sets) with a mean improvement of +5.11% in the noiseless setting and +4.71% under depolarising noise. A decomposed ablation further shows that the full coherent aggregation design outperforms an incoherent baseline on all seven datasets under $p$=0.01 noise (mean +1.80%). Ablations confirm that load balancing is consistently beneficial, while data re-uploading provides the largest gains in the noiseless setting. At top-$k$=1, $W(\varphi)$ serves as a learned post-processing circuit; inter-expert phase interference requires $k \geq 2$. Code is available at https://anonymous.4open.science/r/qmoe-plus/.

## 1 Introduction

The ability to conditionally compute and activate only the model components most relevant to a given input has become a central principle in modern machine learning. Originating with the *adaptive mixture of local experts* (Jacobs et al., 1991), this idea uses a gating network to partition the input space and assign specialised sub-networks. Subsequent work placed this framework on firmer probabilistic foundations (Jordan & Jacobs, 1993) and showed that sparsely activating the top-$k$ experts enables large effective model capacity at low computational cost (Shazeer et al., 2017). This paradigm now underpins large-scale systems such as GShard, Switch Transformer, ST-MoE, and Mixtral (Lepikhin et al., 2021; Fedus et al., 2022; Zoph et al., 2022; Jiang et al., 2024). However, sparse MoE training introduces well-known challenges, including non-differentiable routing, load imbalance leading to expert collapse, and additional optimisation complexity requiring carefully designed auxiliary losses.

In parallel, quantum machine learning (QML) explores whether similar gains can be achieved using quantum systems, leveraging superposition, entanglement, and interference (Biamonte et al., 2017; Cerezo et al., 2021a). In the NISQ era (Preskill, 2018), the dominant framework is the *parameterized quantum circuit* (PQC), a fixed circuit with trainable parameters optimised via a hybrid quantum-classical loop (Benedetti et al., 2019; Mitarai et al., 2018). PQCs can represent expressive function classes through entanglement and *data re-uploading* (Pérez-Salinas et al., 2020; Schuld et al., 2021), but their practical scalability is limited by barren plateaus and hardware noise (McClean et al., 2018; Holmes et al., 2022). These constraints make large monolithic circuits difficult to train, motivating modular architectures that distribute learning across smaller, more tractable quantum components.

The natural intersection of these two lines of work, conditional computation and quantum learning, is the quantum mixture of experts. Several recent contributions have begun to explore this space from different angles. Nguyen et al. (2025) introduce QMoE, to our knowledge the first architecture to realise both the experts and the routing mechanism as quantum circuits: a variational router selects among four parallel

PQC experts via computational-basis measurement, and the selected expert outputs are aggregated classically. Concurrent work by Bhati et al. (2025) employs a *classical* gating network over a heterogeneous pool of quantum experts - including quantum SVMs, QKNNs, QCNNs, and standard QNNs - and reports generalisation improvements on Iris, Titanic, health insurance, and MNIST digit pairs. A different angle is pursued by Heddad & Bouanane (2025), who retain classical experts and use a quantum router to isolate the contribution of quantum interference to the routing decision, observing that wave-interference-based routing achieves superior non-linear separation on the Two Moons benchmark with greater parameter efficiency. Tognini et al. (2025) explore a globally coupled mixture trained end-to-end on MNIST. Together, these works establish that combining quantum circuits with mixture-of-experts structure is both feasible and empirically promising.

A closer reading reveals three shared limitations. First, all prior approaches use single-block PQC experts, which restrict the range of Fourier components they can represent and limit expert specialisation (Pérez-Salinas et al., 2020; Schuld et al., 2021). Second, none include an explicit mechanism for load balancing. As established in classical MoE literature, training without such regularisation tends to concentrate routing on a few experts, leading to collapse (Shazeer et al., 2017; Fedus et al., 2022), and there is no reason to expect quantum models to behave differently. Third, existing aggregation schemes treat routing purely as a selection mechanism: once an expert is chosen, the magnitude of routing amplitudes does not influence the final output. This prevents the model from weighting experts based on confidence. To address this, we introduce a coherent aggregation circuit that operates over the joint routing-data Hilbert space, enabling amplitude-aware expert combination.

**Our Contributions.** We present **QMoE+**, a hybrid quantum mixture of experts that addresses these limitations. Here "hybrid" carries its standard meaning in quantum machine learning: all circuit components are fully quantum, while the training loop (parameter-shift gradients, Adam optimiser) is classical (Benedetti et al., 2019; Cerezo et al., 2021a), distinct from designs pairing classical experts with a quantum router (Heddad & Bouanane, 2025) or a classical gate with quantum experts (Bhati et al., 2025). Our contributions are:

1. **Data re-uploading experts.** We replace single-block PQC experts with two-block DRU experts $U(\mathbf{x}) \to V_1 \to U(\mathbf{x} + \boldsymbol{\phi}) \to V_2$, where $\boldsymbol{\phi}$ is a learnable per-expert offset. This improves expressivity and enables specialisation, yielding the largest accuracy gains in our ablations (Pérez-Salinas et al., 2020).

2. **Coherent aggregation.** We introduce a joint routing-data circuit that entangles routing amplitudes with expert states before measurement. This allows amplitude-weighted expert combination, rather than treating routing as a purely classical selection signal.

3. **Load balancing and sparse routing.** We add load-balancing regularisation to prevent expert collapse, using entropy-based loss for dense routing and Switch-style loss for top-$k$ routing (Fedus et al., 2022). This enables stable training and effective sparsity (e.g., $k=1$, $K=4$).

4. **Comprehensive evaluation.** We evaluate across multiple datasets, gate sets, and both noiseless and noisy ($p=0.01$) settings. QMoE+ consistently outperforms the QMoE baseline (Nguyen et al., 2025), with ablations confirming that each component contributes independently.

The remainder of the paper is organised as follows. Section 2 reviews classical MoE, QML, and prior QMoE work. Section 3 introduces notation and preliminaries. Section 4 presents the QMoE+ architecture, and Section 5 describes the experimental setup. Section 6 reports results under noiseless and noisy conditions, and Section 7 concludes. Additional theoretical analysis and implementation details are provided in Appendices D and E.

## 2 Related Work

The mixture-of-experts framework originates with Jacobs et al. (1991) and Jordan & Jacobs (1993), and was scaled to modern systems through sparse top-$k$ routing with auxiliary load-balancing losses to prevent

expert collapse (Shazeer et al., 2017; Lepikhin et al., 2021; Fedus et al., 2022; Zoph et al., 2022; Jiang et al., 2024). We adopt the Switch Transformer formulation of Fedus et al. (2022) for our sparse routing regime. On the quantum side, parameterized quantum circuits in the NISQ regime (Benedetti et al., 2019; Cerezo et al., 2021b; Bharti et al., 2022) are limited by barren plateaus (McClean et al., 2018; Holmes et al., 2022; Larocca et al., 2025) and by the restricted Fourier spectrum of single-encoding circuits (Schuld et al., 2021), motivating data re-uploading (Pérez-Salinas et al., 2020) as a route to higher expressivity. Three recent works combine these threads. Nguyen et al. (2025) introduce a fully quantum MoE with single-block PQC experts and classical aggregation, which serves as our primary baseline. Bhati et al. (2025) pair a classical router with heterogeneous quantum experts, Heddad & Bouanane (2025) use a quantum router with classical experts, and Tognini et al. (2025) train a globally coupled quantum mixture. None of these jointly addresses multi-block expert expressivity, load balancing, and coherent aggregation. An extended discussion appears in Appendix A.

## 3 Background

### 3.1 Classical Mixture of Experts

A mixture of experts (MoE) decomposes a complex learning task across $K$ specialised sub-networks. Formally, given an input $\mathbf{x} \in \mathbb{R}^d$, a gating network $\mathcal{G}(\mathbf{x}; \psi) = [g_1(\mathbf{x}), \ldots, g_K(\mathbf{x})]$ produces a probability distribution over the $K$ experts, where $g_k(\mathbf{x}) \geq 0$ and $\sum_k g_k(\mathbf{x}) = 1$. Each expert $E_k(\mathbf{x}; \omega_k)$ produces a prediction, and the combined output is

$$\mathbf{y}(\mathbf{x}) = \sum_{k=1}^{K} g_k(\mathbf{x}) E_k(\mathbf{x}; \omega_k). \tag{1}$$

Gating network and experts are trained jointly via gradient descent, enabling the gate to learn which expert is most suited to each region of the input space while experts simultaneously specialise to their assigned regions (Jacobs et al., 1991; Jordan & Jacobs, 1993).

In modern large-scale systems, the dense sum of Eq. (1) is replaced by *sparse* gating: only the top-$k$ experts (with $k \ll K$) are activated per input, reducing forward-pass cost while preserving overall model capacity (Shazeer et al., 2017; Fedus et al., 2022). Because top-$k$ selection is non-differentiable, training typically uses the straight-through estimator (Bengio et al., 2013) or Gumbel-softmax relaxations (Jang et al., 2017; Maddison et al., 2017) to pass gradients through the discrete routing decision.

A persistent challenge in MoE training is *expert collapse*: the gate concentrates routing weight on a small subset of experts, leaving the remainder without meaningful gradient signal and effectively wasted. This is a stable failure mode whenever the routing and task losses are coupled - whichever expert is marginally better early in training receives more data, sharpens its advantage, and eventually monopolises the gate. To counteract this, MoE systems universally include an auxiliary *load-balancing loss* that penalises uneven expert utilisation (Shazeer et al., 2017; Fedus et al., 2022; Zoph et al., 2022). The Switch Transformer formulation (Fedus et al., 2022), which we adopt for our sparse routing regime, computes

$$\mathcal{L}_{\text{load}} = K \sum_{k=1}^{K} f_k \cdot P_k, \tag{2}$$

where $f_k$ is the fraction of the current batch dispatched to expert $k$ (stop-gradient) and $P_k$ is the mean routing probability for expert $k$ (has gradient). Minimising Eq. (2) drives gradient into the routing parameters in proportion to over-utilisation without requiring a differentiable dispatch function.

### 3.2 Parameterized Quantum Circuits

An $n$-qubit Parameterized quantum circuit (PQC) consists of a data-encoding unitary $U(\mathbf{x})$, a parameterized variational unitary $V(\boldsymbol{\theta})$, and a final measurement of a Hermitian observable $\mathcal{O}$. The circuit expectation is

$$f(\mathbf{x}, \boldsymbol{\theta}) = \langle \mathcal{O} \rangle_{\mathbf{x}, \boldsymbol{\theta}} = \langle 0^{\otimes n} | U^\dagger(\mathbf{x}) V^\dagger(\boldsymbol{\theta}) \mathcal{O} V(\boldsymbol{\theta}) U(\mathbf{x}) | 0^{\otimes n} \rangle. \tag{3}$$

Parameters $\boldsymbol{\theta}$ are optimised through a hybrid quantum-classical loop using parameter-shift gradient rules (Mitarai et al., 2018) or automatic differentiation frameworks (Wang et al., 2022a). The choice of encoding $U(\mathbf{x})$ and variational ansatz $V(\boldsymbol{\theta})$ jointly determine the expressivity and trainability of the circuit (Benedetti et al., 2019; Cerezo et al., 2021a).

**Encoding.** We use *angle encoding* (phase encoding) with a two-gate, multi-block structure. For an $n$-qubit register and input $\mathbf{x} \in \mathbb{R}^d$ with $d = 2nB$ for integer $B$, the encoding is applied in $B$ successive blocks. In block $b$, feature pair $(x_{b \cdot 2n+2q}, x_{b \cdot 2n+2q+1})$ is encoded on qubit $q$ as $R_X(x_{b \cdot 2n+2q}) R_Y(x_{b \cdot 2n+2q+1})$, followed by a CNOT entanglement ring $q \to (q + 1) \bmod n$:

$$U(\mathbf{x}) = \prod_{b=0}^{B-1} \left[ \text{CNOT}_{\text{ring}} \cdot \bigotimes_{q=0}^{n-1} R_X(x_{b,q,0}) R_Y(x_{b,q,1}) \right]. \tag{4}$$

This structure ensures that each feature is encoded once per block, and the CNOT ring introduces entanglement between qubits after each encoding step.

**Variational ansatz.** Each variational layer applies parameterized rotations followed by a CNOT ring. The rotation gate set is one of $\{R_X, R_Y, R_X R_Y, R_X R_Y R_Z\}$, where the choice determines the number of parameters per qubit per layer (1, 1, 2, or 3 respectively). With $L$ variational layers and $n$ qubits the total variational parameter count per circuit block is $L \cdot n \cdot |\mathcal{S}|$, where $|\mathcal{S}| \in \{1, 2, 3\}$ is the gate set size.

**Measurement.** Class logits are extracted as Pauli-$Z$ expectation values on the first $C$ qubits (where $C$ is the number of classes):

$$\ell_j = \langle Z_j \rangle = \sum_{s \in \{0,1\}^n} |\psi_s|^2 (-1)^{s_j}, \qquad j = 0, \ldots, C - 1, \tag{5}$$

where $|\psi_s|^2$ is the Born-rule probability of computational basis state $s$ and $s_j$ denotes its $j$-th bit.

**Barren plateaus.** A well-known trainability obstacle for PQCs is the *barren plateau* (McClean et al., 2018): for sufficiently deep or expressive circuits the variance of the cost gradient vanishes exponentially with system size, making gradient-based optimisation infeasible. This problem is exacerbated by high ansatz expressibility (Holmes et al., 2022) and by hardware noise, which induces a separate, noise-driven form of exponential gradient concentration (Wang et al., 2022b; Larocca et al., 2025). Modular architectures that keep each constituent circuit shallow - such as the expert circuits in our work - partially mitigate barren plateaus by ensuring that no single circuit is deep enough to enter the plateau regime.

### 3.3 Data Re-uploading

Data re-uploading (DRU) (Pérez-Salinas et al., 2020) is the technique of interleaving the encoding unitary with trainable variational blocks, rather than applying encoding only once. A two-block DRU circuit takes the form

$$|\psi(\mathbf{x}, \boldsymbol{\theta}^1, \boldsymbol{\theta}^2)\rangle = V_2(\boldsymbol{\theta}^2) U(\mathbf{x}) V_1(\boldsymbol{\theta}^1) U(\mathbf{x}) |0\rangle^{\otimes n}. \tag{6}$$

The motivation for DRU is formal: Schuld et al. (2021) show that the output of a PQC can be written as a truncated Fourier series in $\mathbf{x}$, whose accessible frequencies are determined solely by the data-encoding gates. A single encoding layer of Pauli rotations gives access to only first-order harmonics; each additional encoding layer expands the frequency spectrum. Consequently, a DRU circuit with $L$ encoding blocks can represent functions of substantially higher frequency than a single-encoding PQC with $L$ variational layers, making DRU a theoretically motivated approach to increasing expressivity without increasing the number of trainable parameters (Pérez-Salinas et al., 2020; Goto et al., 2021).

### 3.4 The QMoE Framework

The QMoE of Nguyen et al. (2025) is, to our knowledge, the first architecture in which both routing and experts are implemented as parameterized quantum circuits. We describe it in our notation, as it serves as the primary baseline.

The model uses a routing register ($n_R$ qubits, $K=2^{n_R}$ experts) and a data register ($n_D$ qubits). Input $\mathbf{x} \in \mathbb{R}^d$ is encoded into both via $U(\mathbf{x})$. A routing circuit $G(\boldsymbol{\theta}_G)$ is applied and measured to obtain probabilities $p_k = |\langle k|G(\boldsymbol{\theta}_G)U(\mathbf{x})|0\rangle|^2$. These act as classical weights: controlled operations apply each expert $E_k(\boldsymbol{\theta}_k)$ to the data register, followed by a fixed aggregation layer and terminal $Z$-basis measurement for logits.

The model output is:

$$\mathbf{y}(\mathbf{x}) \;=\; \text{Measure}\left(\text{Agg}\left(\sum_{k=1}^{K} p_k(\mathbf{x})\, E_k(\boldsymbol{\theta}_k)\, U(\mathbf{x})\, |0\rangle^{\otimes n_D}\right)\right), \tag{7}$$

Training is performed end-to-end via parameter-shift gradients.

Three properties are central. First, each expert is a single-block PQC, limiting expressivity to first-order harmonics (Schuld et al., 2021). Second, routing probabilities are classical, and amplitude information is discarded. Third, aggregation is fixed and non-learnable. These design choices define the limitations addressed by QMoE+ (Section 4).

## 4 Methodology

QMoE+ is a hybrid quantum mixture of experts in which routing, expert computation, and aggregation are all parameterised quantum circuits trained end-to-end. It extends the QMoE framework of Nguyen et al. (2025) through three targeted architectural changes: data re-uploading inside each expert, a low-dimensional routing input with coherent amplitude-based aggregation, and an explicit load-balancing auxiliary loss. We describe each component in turn using the notation established in Section 3, then state the composite training objective. Formal theoretical justifications are deferred to Appendix D; physical realisability on NISQ hardware is discussed in Appendix E.

### 4.1 Data Re-Uploading Experts

Each of the $K=4$ experts $E_k$ is a two-block DRU circuit on $n_D=4$ data qubits. Starting from $|0\rangle^{\otimes n_D}$, the circuit applies phase encoding $U(\mathbf{x})$, followed by a variational block $V_k^{(1)}(\boldsymbol{\theta}_k^{(1)})$, a second encoding of the shifted input $U(\mathbf{x} + \boldsymbol{\phi}_k)$, and a second variational block $V_k^{(2)}(\boldsymbol{\theta}_k^{(2)})$:

$$|\psi_k(\mathbf{x})\rangle \;=\; V_k^{(2)}(\boldsymbol{\theta}_k^{(2)})\, U(\mathbf{x} + \boldsymbol{\phi}_k)\, V_k^{(1)}(\boldsymbol{\theta}_k^{(1)})\, U(\mathbf{x})\, |0\rangle^{\otimes n_D}, \tag{8}$$

where $\boldsymbol{\phi}_k \in \mathbb{R}^d$ is a learnable per-expert re-upload offset, initialised at $\mathbf{0}$. Each variational block consists of $L=2$ layers of single-qubit rotations from gate set $\mathcal{S} \in \{\text{RX, RY, RX+RY, RX+RY+RZ}\}$ followed by a CNOT ring.

The offset $\boldsymbol{\phi}_k$ drives expert specialisation: although experts are identical at initialisation, training induces divergence, giving each expert a distinct input view. The second encoding shifts the Fourier spectrum accessible to each expert (Appendix D.1).

Logits are obtained as $Z$-expectation values on the first $C$ data qubits (Eq. (5)). The parameter count per expert under RX+RY+RZ is $2 \cdot L \cdot n_D \cdot 3 + d = 2 \cdot 2 \cdot 4 \cdot 3 + 64 = 112$.

### 4.2 Routing Circuit

The routing circuit $G(\boldsymbol{\theta}_G)$ operates on $n_R=2$ qubits ($K=4$ experts). It begins with a Hadamard layer, followed by phase encoding of a low-dimensional input projection $\mathbf{x}_{1:8}$ and $L=2$ variational layers:

$$|\alpha(\mathbf{x})\rangle \;=\; V_G(\boldsymbol{\theta}_G)\, U(\mathbf{x}_{1:8})\, H^{\otimes n_R}\, |0\rangle^{\otimes n_R}, \tag{9}$$

yielding a complex amplitude vector $\boldsymbol{\alpha}(\mathbf{x}) \in \mathbb{C}^K$, which is passed coherently to aggregation without measurement.

Routing uses only 8 features to avoid capacity imbalance: encoding all 64 features would require 32 encoding blocks on 2 qubits, overwhelming the 2-layer variational circuit and biasing routing toward encoding

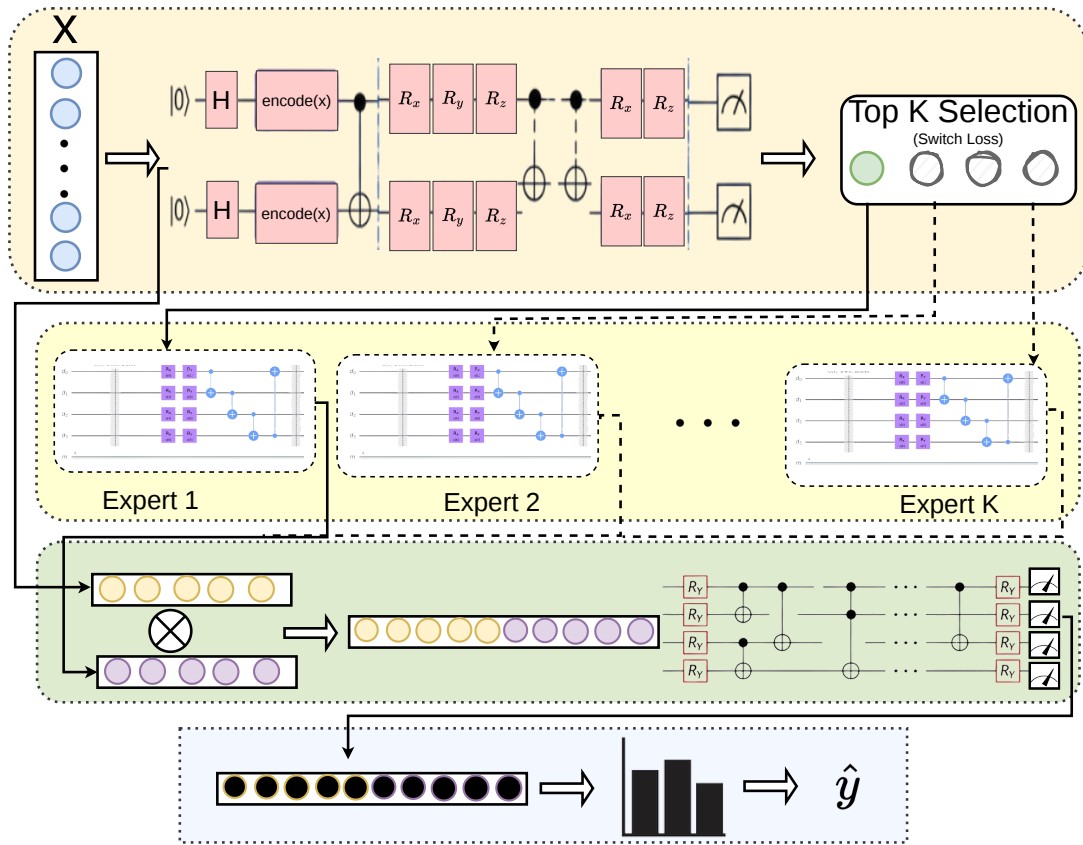

Figure 1: Overview of QMoE+. The routing circuit $G(\boldsymbol{\theta}_G)$ encodes $\mathbf{x}$ on $n_R$ qubits (Hadamard + phase encoding + $L$ variational layers) to produce probabilities $p_k=|\alpha_k|^2$. Top-$k$=1 selects expert $k^*$, with Switch load-balancing $\mathcal{L}_{\text{load}}=K\sum_k f_k P_k$ ($\lambda$=0.01) preventing collapse. Only $k^*$ is evaluated. Each expert is a two-block DRU circuit $U(\mathbf{x}) \to V_1 \to U(\mathbf{x}+\boldsymbol{\phi}) \to V_2$, with learnable offset $\boldsymbol{\phi}$ enabling specialisation. The joint state $\tilde{\alpha}_{k^*}|k^*\rangle_R \otimes |e_{k^*}\rangle_D$ is processed by a coherent aggregation circuit $W(\boldsymbol{\varphi})$ ($R_Y$ + CNOT, $L_{\text{agg}}$=2), and $Z$-measurements on data qubits produce logits, followed by softmax outputs $\hat{y}$.

geometry. Restricting to 8 features yields one encoding block per variational layer, maintaining a balanced expressivity budget.

### 4.3 Coherent Aggregation

Given the routing state $|\alpha(\mathbf{x})\rangle = \sum_{k=0}^{K-1} \alpha_k(\mathbf{x})|k\rangle_R$ and the $K$ expert states $\{|\psi_k(\mathbf{x})\rangle\}_k$, we form the joint routing-data state

$$|\Psi(\mathbf{x})\rangle = \sum_{k=0}^{K-1} \alpha_k(\mathbf{x})|k\rangle_R \otimes |\psi_k(\mathbf{x})\rangle_D \qquad (10)$$

on $n_R + n_D = 6$ total qubits. A learnable variational circuit $W(\boldsymbol{\varphi})$ is then applied over the entire joint register: $L_{\text{agg}}$=2 layers, each consisting of $R_Y$ rotations on every qubit followed by a CNOT ring with wrap-around. Class logits are extracted as $Z$-expectation values on the data qubits:

$$\text{logit}_j(\mathbf{x}) = \langle\Psi(\mathbf{x})| W^\dagger(\boldsymbol{\varphi}) Z_{n_R+j} W(\boldsymbol{\varphi}) |\Psi(\mathbf{x})\rangle, \quad j=0,\ldots,C-1. \qquad (11)$$

The design differs from the fixed controlled-gate aggregation of Nguyen et al. (2025) in two respects. First, $W(\boldsymbol{\varphi})$ acts on all $n_R + n_D$ qubits jointly, including both the routing and data registers, rather than only

routing-to-data controlled operations; this means the learned transformation can couple the routing subspace to the data subspace. Second, $W(\boldsymbol{\varphi})$ is variational and trained end-to-end, so it can adapt the aggregation to the data distribution rather than applying a fixed circuit structure. The routing amplitudes $\alpha_k(\mathbf{x})$ are carried as complex values into the joint state and are not collapsed by intermediate measurement before aggregation; the single terminal measurement on the data register is the only point at which quantum information is converted to classical output. The structural motivation for this design - that discarding the routing state by measurement before aggregation loses the amplitude magnitudes as a conditioning signal - is discussed further in Appendix D.3. The joint-state construction of instantiates the PREPARE-SELECT primitive of the linear combination of unitaries framework (Childs & Wiebe, 2012); our contribution is the application of this structure to mixture-of-experts aggregation and the addition of a learnable variational circuit $W(\varphi)$ acting on the joint routing-data register downstream of the state preparation.

### 4.4 Load-Balancing Regularisation

Let $p_k(\mathbf{x}) = |\alpha_k(\mathbf{x})|^2$ denote the routing probability for expert $k$. Let $\bar{e}_k = \mathbb{E}_{\mathbf{x}\in\mathcal{B}}[p_k(\mathbf{x})]$ denote the mean routing probability for expert $k$ over the current batch $\mathcal{B}$. For dense routing, we use a negative-entropy penalty:

$$\mathcal{L}_{\text{load}}^{\text{dense}} = -H(\bar{\mathbf{e}}) = \sum_{k=0}^{K-1} \bar{e}_k \log \bar{e}_k, \tag{12}$$

which encourages uniform expert utilisation and mitigates collapse (Shazeer et al., 2017; Fedus et al., 2022).

For sparse ($k{=}1$) routing, we adopt the Switch Transformer loss:

$$\mathcal{L}_{\text{load}}^{\text{sparse}} = K \sum_{k=0}^{K-1} f_k \cdot P_k, \tag{13}$$

where $f_k$ is the fraction of samples routed to expert $k$ (stop-gradient), and $P_k = \bar{e}_k$ retains gradient. This provides a stable signal without requiring differentiable dispatch.

The full objective is:

$$\mathcal{L} = \mathcal{L}_{\text{CE}} + \lambda(t)\,\mathcal{L}_{\text{load}}, \tag{14}$$

with a linear warm-up:

$$\lambda(t) = \begin{cases} 0 & t < t_{\text{start}} \\ \lambda_{\text{target}} \cdot \dfrac{t - t_{\text{start}}}{t_{\text{end}} - t_{\text{start}}} & t_{\text{start}} \leq t < t_{\text{end}} \\ \lambda_{\text{target}} & t \geq t_{\text{end}} \end{cases} \tag{15}$$

where $t_{\text{start}}{=}5$, $t_{\text{end}}{=}15$, and $\lambda_{\text{target}}{=}0.01$. This avoids early over-regularisation before experts specialise (Fedus et al., 2022; Zoph et al., 2022).

The load loss depends only on $p_k = |\alpha_k|^2$, so gradients affect routing parameters $\boldsymbol{\theta}_G$ only, leaving expert and aggregation parameters unchanged.

### 4.5 Sparse Top-$k$ Routing

QMoE+ uses top-$k{=}1$ sparse routing with $K{=}4$ experts. For each input, routing amplitudes are computed, the top expert is selected via $\arg\max_k |\alpha_k|^2$, and the remaining amplitudes are zeroed and renormalised. Only the selected expert is evaluated per sample, while the others are skipped. With load balancing (Eq. (13)), each expert processes approximately one quarter of the batch, yielding a $4\times$ reduction in computation relative to dense routing without loss in accuracy.

Sparsification is applied in amplitude space prior to the joint state construction (Eq. (10)). The sparse vector $\tilde{\boldsymbol{\alpha}}$ retains the top-$k$ entries of $\boldsymbol{\alpha}$ and is renormalised, ensuring that the aggregation circuit $W(\boldsymbol{\varphi})$ operates on a valid normalised state and preserves coherence.

---

**Algorithm 1** QMoE+ Forward Pass (sparse top-$k$=1 variant)

---

**Require:** $\mathbf{x} \in [0, \pi]^d$; $\{\boldsymbol{\theta}_k^{(1)}, \boldsymbol{\theta}_k^{(2)}, \boldsymbol{\phi}_k\}_{k=0}^{K-1}$; $\boldsymbol{\theta}_G$; $\boldsymbol{\varphi}$
**Ensure:** Logits $\mathbf{l} \in \mathbb{R}^C$
 1: **Routing:** $|\alpha(\mathbf{x})\rangle \leftarrow V_G(\boldsymbol{\theta}_G) \, U(\mathbf{x}_{1:8}) \, H^{\otimes n_R} \, |0\rangle^{\otimes n_R}$ ▷ Eq. (9)
 2: $k^* \leftarrow \arg\max_k |\alpha_k(\mathbf{x})|^2$; $\tilde{\alpha}_k \leftarrow \alpha_k \cdot \mathbf{1}[k = k^*] \, / \, |\alpha_{k^*}|$
 3: **Expert evaluation:** $|\psi_k(\mathbf{x})\rangle \leftarrow V_k^{(2)} \, U(\mathbf{x}+\boldsymbol{\phi}_k) \, V_k^{(1)} \, U(\mathbf{x}) \, |0\rangle^{\otimes n_D}$ for $k = k^*$ ▷ Eq. (8) (set $|\psi_k\rangle = \mathbf{0}$ for $k \neq k^*$)
 4: **Joint state:** $|\Psi(\mathbf{x})\rangle \leftarrow \sum_k \tilde{\alpha}_k \, |k\rangle_R \otimes |\psi_k(\mathbf{x})\rangle_D$ ▷ Eq. (10)
 5: **Aggregation:** $\mathbf{l} \leftarrow \left[ \langle Z_{n_R+j} \rangle_{W(\boldsymbol{\varphi})|\Psi\rangle} \right]_{j=0}^{C-1}$ ▷ Eq. (11)
 6: **return** $\mathbf{l}$

---

### 4.6 Complete Architecture

Figure 1 summarises the complete QMoE+ forward pass. The architecture has the following parameter count in the RX+RY+RZ configuration:

- **Routing circuit**: $L \cdot n_R \cdot |\mathcal{S}| = 2 \cdot 2 \cdot 3 = 12$ parameters.

- **Each DRU expert**: $2 \cdot L \cdot n_D \cdot |\mathcal{S}| + d = 48 + 64 = 112$ parameters.

- **Aggregation circuit**: $L_{\text{agg}} \cdot (n_R + n_D) \cdot 1 = 2 \cdot 6 \cdot 1 = 12$ parameters (RY only).

- **Total**: $12 + 4 \times 112 + 12 = \mathbf{472}$ parameters.

All parameters are optimised jointly with the Adam optimiser (Kingma & Ba, 2015) at learning rate $2 \times 10^{-3}$. Expert parameters $\boldsymbol{\theta}_k^{(1)}, \boldsymbol{\theta}_k^{(2)}$ are initialised from $\mathcal{N}(0, 1/\sqrt{n_{\text{par}}})$; re-upload offsets $\boldsymbol{\phi}_k$ are initialised to $\mathbf{0}$; aggregation parameters $\boldsymbol{\varphi}$ are initialised from $\mathcal{N}(0, 0.01)$ to keep the initial aggregation circuit close to the identity.

## 5 Experimental Details

### 5.1 Datasets

We evaluate on seven classification benchmarks spanning image, synthetic, and tabular data.

**Image datasets.** We use MNIST (LeCun et al., 2010) and Fashion-MNIST (Xiao et al., 2017) in binary (2-class) and four-class settings. Images are resized from 28×28 to 8×8, flattened to 64 dimensions, and scaled to $[0, \pi]$ for angle encoding. Up to 6,000 training and 1,000 test samples are used per class, with labels remapped to $\{0, \ldots, C-1\}$.

**Synthetic dataset.** We use a mixture-of-functions binary dataset (Pérez-Salinas et al., 2020) with 5,000 samples and 8 input features, scaled to $[0, \pi]$.

**Tabular datasets.** We evaluate on UCI Wine (Cortez et al., 2009) (3 classes, 13 features, padded to 16) and Breast Cancer (Wolberg et al., 1994) (2 classes, 30 features, padded to 32). Features are MinMax-scaled to $[0, \pi]$ using statistics from the training split.

### 5.2 Models and Baselines

We compare four model configurations across all datasets and gate sets. The same encoding, gate sets, optimiser, and preprocessing are used throughout.

- **Single PQC.** A single-block PQC with one phase-encoding layer followed by $L{=}2$ variational layers: $V(\boldsymbol{\theta})\,U(\mathbf{x})\,|0\rangle^{\otimes n_D}$. No DRU, no MoE. This is the direct analogue of the quantum baseline of Nguyen et al. (2025).

- **DRU Only.** A single two-block DRU circuit, Eq. (8), with one learnable re-upload offset $\phi$. No MoE. This isolates the contribution of data re-uploading from that of the mixture-of-experts structure.

- **QMoE Baseline.** Our re-implementation of Nguyen et al. (2025): $K{=}4$ single-block PQC experts. The QMoE Baseline corresponds to top-$k{=}4$ (all experts active, dense routing), matching the $K{=}4$ configuration reported by Nguyen et al. (2025).

- **QMoE+ (ours).** The full architecture of Section 4: $K{=}4$ DRU experts, coherent routing on $\mathbf{x}_{1:8}$, coherent aggregation on the 6-qubit joint register, and $\mathcal{L}_{\text{load}}$ via Eq. (13). **Top-$k{=}1$ sparse routing** is the primary configuration: each input is routed to exactly one expert, and the load-balancing loss ensures near-uniform dispatch across experts. This is the model reported in the main results tables.

Gate sets vary across four configurations: RX, RY, RX+RY, and RX+RY+RZ, controlling the rotation gates in all variational layers identically across all models. The data encoding always uses both $R_X$ and $R_Y$ as specified in Eq. (4), independent of the gate set. All model implementations use TorchQuantum (Wang et al., 2022a). We include comparison with QMoE (Nguyen et al., 2025), which we re-implement as a baseline. We do not include direct empirical comparisons with other quantum MoE works (Bhati et al., 2025; Heddad & Bouanane, 2025; Tognini et al., 2025), as their implementations are difficult to reproduce and in several cases rely on partially classical routing or expert components, making controlled comparisons inconsistent with our fully quantum setting.

## 5.3 Training Protocol

All models are trained with the Adam optimiser (Kingma & Ba, 2015) at learning rate $2{\times}10^{-3}$ and batch size 32, for up to 50 epochs. Early stopping is applied with patience 10 epochs on exact (noiseless) validation accuracy, with minimum improvement threshold $10^{-4}$. The load-balancing coefficient $\lambda(t)$ follows the warm-up schedule of Eq. (15) with $t_{\text{start}}{=}5$, $t_{\text{end}}{=}15$, $\lambda_{\text{target}}{=}0.01$.

All runs are replicated over 5 independent random seeds. Results are reported as the mean $\pm$ standard deviation over the 5 seeds.

## 5.4 Evaluation Protocol

**Noiseless setting.** Training and early stopping use exact statevector simulation without shot noise. Checkpoint selection is based exclusively on validation accuracy: all datasets are split into training and validation sets using an 80/20 stratified partition, and the checkpoint achieving the highest validation accuracy during training is retained. The held-out test set is evaluated only once, at reporting time.

**Noisy setting.** To assess robustness under NISQ conditions, we simulate hardware noise using a depolarising channel with error rate $p{=}0.01$ (Nielsen & Chuang, 2010; Bharti et al., 2022). Noise is injected after each CNOT-ring layer across all circuit blocks. For each qubit, a Pauli error $\{X, Y, Z\}$ is applied with probability $p/3$.

During evaluation, we additionally simulate finite sampling by drawing 1,024 measurement shots from the output distribution. Training remains noise-free and uses exact expectations to preserve stable gradients, while noise and sampling are applied only at evaluation time.

Detailed algorithm for the noiseless and noisy forward passes is provided in Algorithms 2 and 3 in Appendix B.

## 5.5 Ablation Studies

We summarise the ablation settings below; full results are provided in Appendix C.

Table 1: Ablation settings

| Category | Description |
|---|---|
| **Component** | $-$DRU, $-$CoherentAgg, $-$Switch Loss. |
| **Top-$k$** | $k \in \{1, 2, 3, 4\}$ |
| **Heterogeneous** | QMoE+Hetero with QCNN, QSVM, QKNN, and QNN experts. |

All experiments are conducted on NVIDIA GeForce RTX 2080 Ti GPUs (11 GB VRAM). Full implementation details, including parameter counts and per-model hyperparameter tables, are provided in Appendix B.

## 6 Results and Discussion

Tables 2 and 3 report test accuracy (mean $\pm$ std over 5 seeds) for all four models across all seven datasets and four gate sets in the noiseless and $p$=0.01 depolarising-noise settings respectively. QMoE Baseline uses top-$k$=4 (all four experts active, dense routing) as in Nguyen et al. (2025); QMoE+ uses top-$k$=1 (one active expert per input) and therefore evaluates only one expert circuit per forward pass. Despite activating only one quarter of the expert circuits at inference, QMoE+ has substantially higher total capacity (472 vs. 164 parameters for the QMoE Baseline, RX+RY+RZ). Under RX+RY+RZ, QMoE+ activates only 136 parameters per sample (12 routing + 112 expert + 12 aggregation), fewer than the Baseline's 164, despite a 3× larger total budget. This advantage is gate-set-dependent: the per-expert offset $\phi_k \in \mathbb{R}^d$ ($d$=64) is gate-set-independent, so under RX and RY QMoE+ activates 96 versus the Baseline's 68; under RX+RY both activate 116 (Table 4).

### 6.1 Noiseless Results

In the noiseless setting, QMoE+ achieves the highest accuracy in 27 of 28 dataset-gate configurations, with mean gains of +5.11% over the QMoE Baseline and +3.54% over a single DRU expert. These improvements are achieved with top-$k$=1 routing, requiring only one active expert per sample. Gains are largest on more complex tasks, such as Wine (up to +23.34%) and multi-class image classification, where limited expressivity of single-block PQCs becomes a bottleneck (Schuld et al., 2021).

Ablations confirm that DRU experts are the primary driver of accuracy gains relative to the single-block QMoE Baseline, outperforming it on 6 of 7 noiseless datasets by up to +7.3% (Wine; Appendix C.6). Within the full QMoE+ architecture, this contribution is dataset-dependent: on MNIST, DRU's additional circuit depth can marginally reduce accuracy (Table 13, MNIST-2: $-6.4\%$), while on tabular and Fashion datasets it is consistently positive. Coherent aggregation and load balancing add a smaller but reliable top-up, strongest on tabular tasks (Wine +8.9%, Breast Cancer +3.1%). Performance improves with richer gate sets, with RX+RY+RZ performing best overall, while larger relative gains appear under simpler gates where baseline capacity is most constrained.

### 6.2 Noisy Results

Under depolarising noise ($p$=0.01, 1,024 shots), QMoE+ outperforms the QMoE Baseline in 20/28 configurations (mean +4.71%) and DRU-only in 21/28 (mean +1.89%). The average accuracy drop from the noiseless setting is 3.17% for QMoE+ versus 2.77% for the Baseline, reflecting DRU's deeper circuits accumulating more depolarising error. The Baseline matches or exceeds QMoE+ in 8 of 28 configurations, following two patterns: on MNIST-2 and MNIST-4 under multi-axis gate sets (RX+RY, RX+RY+RZ), DRU's doubled CNOT-ring traversal amplifies noise accumulation per expert pass; on Breast Cancer under simple gate sets (RX, RY), both models are near ceiling (87-90%) and the marginal expressivity gain from DRU is smaller than its noise cost-a saturation effect (Appendix C.1). Across Fashion and tabular datasets, QMoE+ maintains consistent gains.

Noise affects richer gate sets more strongly: RX+RY+RZ and RX+RY show the largest degradation due to increased circuit depth, while simpler gates (RX, RY) drop less ($1-3\%$). Additional results (Appendix C.1)

Table 2: Noiseless test accuracy (%, mean ± std over 5 seeds). **QMoE Baseline**: re-implementation of Nguyen et al. (2025), top-$k$=4 (dense, all experts active). **QMoE+**: our model, top-$k$=1 (one expert active per input). **Bold**: highest; underline: second highest per row. Tabular datasets (Wine, Breast Cancer) evaluated by 5-fold cross-validation.

| Dataset | Gate Set | Single PQC | DRU Only | QMoE Baseline ($k$=4) | QMoE+ ($k$=1, ours) |
|---|---|---|---|---|---|
| MNIST-2 | RX | 72.18±1.66 | 76.96±0.96 | 77.85±1.03 | **78.65**±1.38 |
| | RY | 71.00±0.55 | 77.65±1.00 | 78.28±1.22 | **79.92**±2.46 |
| | RX+RY | 75.08±1.78 | 79.57±0.95 | 81.73±0.54 | **82.83**±2.47 |
| | RX+RY+RZ | 78.86±0.46 | 79.96±1.13 | 84.01±1.57 | **85.71**±4.91 |
| MNIST-4 | RX | 40.33±1.43 | 53.92±1.33 | 45.77±0.81 | **56.53**±0.62 |
| | RY | 41.31±1.84 | 51.92±0.15 | 46.55±1.26 | **55.68**±2.23 |
| | RX+RY | 49.61±1.04 | 55.41±1.10 | 56.52±4.68 | **58.08**±1.17 |
| | RX+RY+RZ | 52.41±1.03 | 56.03±1.82 | 58.70±2.12 | **60.91**±5.25 |
| Fashion-2 | RX | 68.02±1.37 | 76.43±0.82 | 74.13±0.69 | **79.78**±0.79 |
| | RY | 66.10±1.18 | 77.02±1.18 | 73.45±1.67 | **78.73**±0.58 |
| | RX+RY | 73.22±0.40 | 78.35±1.05 | 80.18±1.19 | **81.07**±0.91 |
| | RX+RY+RZ | 74.87±0.41 | 78.51±1.01 | 79.66±1.27 | **82.05**±1.80 |
| Fashion-4 | RX | 44.31±2.51 | 56.39±0.67 | 54.06±1.20 | **63.06**±0.86 |
| | RY | 39.58±1.99 | 55.86±1.63 | 51.82±1.39 | **63.52**±1.21 |
| | RX+RY | 47.57±1.85 | 58.46±1.16 | 63.73±1.43 | **63.42**±1.06 |
| | RX+RY+RZ | 51.98±1.55 | 59.96±0.74 | 63.96±2.59 | **65.51**±2.11 |
| Synthetic | RX | 54.62±3.58 | 70.70±3.38 | 61.04±1.98 | **73.94**±1.66 |
| | RY | 66.06±1.19 | 69.58±2.76 | 66.22±1.23 | **71.52**±1.71 |
| | RX+RY | 67.44±1.23 | 71.52±2.78 | 69.30±1.03 | **72.70**±1.12 |
| | RX+RY+RZ | 68.12±1.05 | 72.54±1.90 | 70.68±1.39 | **75.49**±2.89 |
| Wine | RX | 35.93±16.49 | 53.33±13.82 | 34.44±14.26 | **57.78**±4.79 |
| | RY | 38.52±5.91 | 51.11±23.23 | 34.07±4.06 | **52.96**±1.92 |
| | RX+RY | 48.89±13.33 | 52.59±11.54 | 60.30±11.67 | **61.85**±3.84 |
| | RX+RY+RZ | 56.30±8.94 | 64.81±9.35 | 67.78±15.96 | **69.28**±1.60 |
| Breast Cancer | RX | 82.44±10.46 | 89.47±4.02 | 92.11±2.06 | **92.23**±3.16 |
| | RY | 83.33±11.80 | 91.58±1.47 | 91.75±2.11 | **92.85**±3.95 |
| | RX+RY | 84.54±7.53 | 92.63±2.88 | 91.63±2.88 | **93.98**±2.56 |
| | RX+RY+RZ | 90.00±5.29 | 92.63±2.95 | 91.11±2.91 | **94.02**±3.05 |

show near-noiseless performance at $p$=0.001 and severe degradation at $p$=0.05. Overall, $p$=0.01 represents a practical regime, highlighting the trade-off between expressivity and noise robustness.

**Scope of the noise model.** These results use the modular depolarising protocol of Algorithm 3 in B, in which noise is injected after every CNOT-ring layer of each component independently- the standard NISQ simulation practice (Cerezo et al., 2021a; Wang et al., 2022a; Bharti et al., 2022). They should *not* be read as predictions for the hardware-equivalent PREPARE-SELECT circuit (Appendix E), whose additional controlled-gate overhead yields ≈91% state fidelity at $p$=0.001 but only ≈62% at $p$=0.01 (Table 8); the practical deployment regime for the full coherent architecture is therefore $p \lesssim 0.001$. See Appendix E, Table 24 for the full two-qubit gate count and fidelity analysis of the hardware-equivalent PREPARE-SELECT circuit.

# 7 Conclusion and Future Work

We presented QMoE+, a hybrid quantum mixture of experts that improves on the QMoE framework of Nguyen et al. (2025) through three targeted architectural changes: two-block DRU experts with learnable re-upload offsets, a coherent aggregation circuit over the joint routing-data Hilbert space, and a Switch Transformer load-balance loss. Across seven datasets, four gate sets, and two noise levels, QMoE+ consistently outperforms both the QMoE baseline and a single DRU expert in the noiseless setting, with the advantage sustained under $p$=0.01 depolarising noise. The DRU expert modification accounts for the dominant share of the accuracy gain; load-balancing is the most universally reliable contribution; and coherent aggre-

Table 3: Test accuracy (%, mean ± std) under depolarising noise $p$=0.01 applied after every CNOT-ring layer, with 1,024-shot readout noise at evaluation. Same model configuration as Table 2. **Bold**: highest; underline: second highest per row.

| Dataset | Gate Set | Single PQC | DRU Only | QMoE Baseline ($k$=4) | QMoE+ ($k$=1, ours) |
|---|---|---|---|---|---|
| MNIST-2 | RX | 70.01±1.45 | 75.32±1.72 | 73.46±0.80 | **76.18**±2.04 |
| | RY | 70.57±0.70 | 75.47±1.15 | 73.61±1.03 | **76.13**±2.23 |
| | RX+RY | 71.25±1.73 | 77.77±0.99 | **79.37**±0.48 | 76.40±2.78 |
| | RX+RY+RZ | 75.62±1.10 | 78.61±0.84 | **79.27**±2.04 | 78.90±1.81 |
| MNIST-4 | RX | 39.22±1.55 | **52.49**±0.97 | 45.70±0.84 | 51.46±1.12 |
| | RY | 40.90±1.55 | **51.25**±0.68 | 45.78±0.88 | 51.17±0.91 |
| | RX+RY | 48.14±1.15 | 53.41±1.07 | **57.05**±4.15 | 52.78±0.96 |
| | RX+RY+RZ | 51.66±0.60 | 53.75±1.62 | **59.26**±2.15 | 52.68±0.89 |
| Fashion-2 | RX | 66.50±1.23 | 75.15±1.42 | 60.94±0.40 | **77.73**±1.12 |
| | RY | 65.89±1.19 | 76.12±0.92 | 64.27±1.51 | **77.55**±1.02 |
| | RX+RY | 72.13±0.62 | 77.21±0.88 | 68.04±1.05 | **77.57**±0.80 |
| | RX+RY+RZ | 73.78±0.59 | 77.58±0.95 | 70.03±0.67 | **78.54**±0.63 |
| Fashion-4 | RX | 42.12±2.17 | 54.89±1.03 | 53.18±0.76 | **60.09**±1.38 |
| | RY | 38.45±1.62 | 55.15±0.67 | 51.21±1.20 | **59.85**±1.37 |
| | RX+RY | 46.98±0.94 | 56.88±1.65 | 61.48±1.49 | **61.78**±0.89 |
| | RX+RY+RZ | 50.49±1.32 | 57.70±0.71 | 62.91±2.52 | **63.01**±0.54 |
| Synthetic | RX | 54.26±1.56 | 68.76±1.53 | 60.08±1.95 | **72.14**±1.09 |
| | RY | 64.36±1.04 | 67.38±2.14 | 65.92±1.35 | **70.70**±1.56 |
| | RX+RY | 66.70±0.73 | 70.68±2.29 | 68.04±1.44 | **71.96**±1.36 |
| | RX+RY+RZ | 67.18±1.08 | 70.46±1.72 | 68.10±1.62 | **73.82**±1.06 |
| Wine | RX | 34.26±10.82 | 47.04±21.71 | 32.96±13.49 | **55.56**±4.83 |
| | RY | 36.96±9.76 | 50.74±20.47 | 33.30±10.14 | **51.85**±8.53 |
| | RX+RY | 48.26±12.32 | 50.74±15.70 | **60.22**±9.07 | 58.15±6.79 |
| | RX+RY+RZ | 56.70±8.81 | 66.37±6.73 | **67.81**±8.57 | 67.41±8.57 |
| Breast Cancer | RX | 81.44±8.57 | 87.82±3.53 | **88.11**±2.00 | 87.72±2.88 |
| | RY | 82.33±10.56 | 90.75±1.19 | **90.40**±3.16 | 87.89±3.25 |
| | RX+RY | 83.72±6.86 | 90.98±2.35 | 91.98±2.42 | **93.16**±2.03 |
| | RX+RY+RZ | 89.35±4.40 | 91.81±2.57 | 90.93±2.57 | **92.98**±3.80 |

gation provides consistent gains confirmed by a decomposed ablation isolating the coherence contribution $\Delta_{coh} = (a)-(b)$ from the learnability contribution $\Delta_{learn} = (b)-(c)$, which establishes that the full coherent design outperforms a fixed incoherent baseline in all seven datasets under $p$=0.01 noise (mean +1.80%) and in four of seven noiseless (mean +1.38%). At the top-$k$=1 operating point used for deployment, $W(\varphi)$ acts as a learned quantum post-processing stage; the inter-expert phase interference that the coherent design structurally enables requires $k \geq 2$, where multiple routing amplitudes are simultaneously nonzero.

**Limitations.** Two aspects of the current work invite further investigation. First, the expressivity advantage of DRU experts comes with greater circuit depth, which increases sensitivity to gate noise at rates above $p \approx 0.01$; the appropriate operating regime for QMoE+ is the $10^{-3}$-$10^{-2}$ gate error range characteristic of current NISQ hardware (Bharti et al., 2022). Second, the joint-state construction is evaluated via statevector simulation; the equivalent PREPARE-SELECT circuit for hardware execution carries an additional two-qubit gate overhead that is within reach at $p$=0.001 but remains challenging at $p$=0.01 (Appendix E).

**Future Work.** Several natural extensions follow from this work. Scaling to larger expert pools and deeper routing circuits would test whether the load-balancing and DRU mechanisms generalise beyond the $K$=4 setting studied here. Extending the architecture to tasks beyond classification - such as variational eigensolvers or combinatorial optimisation - would establish the breadth of the re-uploading expressivity benefit. On the hardware side, approximate circuit compilation of the PREPARE-SELECT structure offers a concrete path toward near-term physical deployment. Finally, a routing analysis that tracks expert selection across inputs and training epochs would clarify the degree to which the router learns task-specific specialisation.

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

# Appendix

# A    Related Work

## A.1    Classical Mixture of Experts

The mixture-of-experts framework traces its origins to Jacobs et al. (1991), who showed that a soft gating network can decompose a complex regression problem into specialised sub-tasks by partitioning the input space probabilistically. Jordan & Jacobs (1993) extended this to a recursive, hierarchical structure and provided an expectation-maximisation training procedure that admits a clean probabilistic interpretation. For two decades these ideas remained primarily of theoretical interest; their return to prominence came with Shazeer et al. (2017), who demonstrated that sparse top-$k$ routing - activating only a small subset of experts per input - allows a single model to hold vastly more parameters than a dense network of comparable compute cost. The practical impact was immediate: GShard (Lepikhin et al., 2021) applied sparsely-gated MoE to multilingual translation at scale, Switch Transformer (Fedus et al., 2022) pushed the design to a top-1 rule and showed that a simple, uniform load-balancing auxiliary loss was sufficient to prevent expert collapse, and ST-MoE (Zoph et al., 2022) identified additional instabilities and addressed them through router $z$-loss regularisation. Mixtral (Jiang et al., 2024) subsequently demonstrated that eight-expert MoE layers, with two active per token, achieve state-of-the-art open-weight performance with a fraction of the inference FLOPs of dense competitors.

Two recurring technical challenges are central to this literature and directly inform our design. First, the routing decision is inherently discrete - selecting which experts receive each token - yet must be made differentiable for end-to-end training; approaches include straight-through estimators (Bengio et al., 2013) and continuous relaxations via the Gumbel-softmax (Jang et al., 2017; Maddison et al., 2017). Second, and more relevant to our work, expert collapse is a stable fixed point of gradient descent in the absence of explicit counter-pressure: whichever expert achieves a marginally lower loss early in training receives more routing weight, which in turn sharpens its advantage and starves the others of signal. The auxiliary losses of Shazeer et al. (2017) and Fedus et al. (2022) address this through importance and load-balance penalties respectively; we adopt the Switch Transformer formulation for our sparse routing regime and a negative-entropy penalty for our dense routing ablations, as described in Section 4.

## A.2    Parameterized Quantum Circuits and Quantum Machine Learning

Quantum machine learning in the NISQ era (Preskill, 2018; Bharti et al., 2022) is dominated by parameterized quantum circuits (PQCs) (Benedetti et al., 2019; Cerezo et al., 2021a), hybrid models that encode classical data into quantum states, apply a trainable unitary, and extract predictions via observable expectations. Training proceeds through a classical optimiser using parameter-shift gradient rules (Mitarai et al., 2018) or automatic differentiation frameworks such as TorchQuantum (Wang et al., 2022a). This paradigm has produced quantum analogues of many classical architectures: quantum neural networks (Abbas et al., 2021), quantum convolutional networks (Cong et al., 2019), quantum support vector machines (Havlíček et al., 2019; Rebentrost et al., 2014), and quantum autoencoders (Romero et al., 2017).

A precise understanding of what PQCs can express has been developed through the Fourier analysis of quantum models (Schuld et al., 2021). Because data-encoding gates contribute Pauli-rotation frequencies, a PQC with a single encoding layer can represent only a limited-bandwidth trigonometric polynomial in the input. *Data re-uploading* (DRU) (Pérez-Salinas et al., 2020) overcomes this limitation by interleaving encoding layers with trainable blocks, expanding the accessible frequency spectrum with each repetition; Schuld et al. (2021) show formally that models with sufficiently many re-uploads can realise any set of Fourier coefficients and are therefore universal function approximators. This is a strict theoretical improvement over single-encoding architectures, and it motivates the expert design at the core of our work.

Trainability of PQCs is complicated by the barren plateau phenomenon: for sufficiently expressive or deep circuits, gradients vanish exponentially with system size, making optimisation infeasible (McClean et al., 2018). Critically, this problem is exacerbated by both circuit expressibility (Holmes et al., 2022) and hardware noise (Wang et al., 2022b; Larocca et al., 2025). The latter is particularly relevant here: Larocca et al. (2025) show that depolarising noise induces deterministic exponential concentration of gradients regardless of architecture, which is one reason why testing QML models under realistic noise levels is a meaningful evaluation and not merely an engineering concern. Structured circuit families - quantum convolutional networks (Cong et al., 2019), local-measurement architectures (Cerezo et al., 2021b) - have been proposed as partial remedies. Our approach of distributing learning across small, shallow DRU experts addresses trainability from a different angle: each expert circuit is independently shallow, keeping its local landscape well-conditioned even when the system as a whole would exhibit a plateau if implemented as a single deep circuit.

## A.3   Quantum Mixtures of Experts

The integration of MoE structures into QML is recent and remains largely unexplored. Nguyen et al. (2025) introduce QMoE, the first fully quantum MoE, where both routing and experts are implemented as PQCs. A quantum routing circuit activates multiple experts in superposition, and their outputs are aggregated before measurement. QMoE outperforms single-PQC baselines on MNIST and Fashion-MNIST, demonstrating the promise of quantum MoEs.

However, its design has key limitations. First, experts are restricted to single-block PQCs, limiting expressivity (Schuld et al., 2021). Second, the routing circuit operates on the full encoded input, which can reduce discriminability by reflecting encoding structure rather than task-relevant features. Third, there is no load-balancing objective, leading to potential expert collapse, as observed in classical MoEs.

Subsequent work explores alternative designs. Bhati et al. (2025) use a classical router with heterogeneous quantum experts, improving stability but removing any quantum contribution to routing. Heddad & Bouanane (2025) show that quantum routing alone can provide expressive advantages, though paired with classical experts. Tognini et al. (2025) demonstrate fully quantum, jointly trained mixtures, but do not address expert specialisation or load balancing. Together, these works highlight both the promise of quantum MoEs and the need for improved routing, expert design, and training stability.

## A.4   Positioning of This Work

Prior quantum MoE work leaves three key gaps. First, experts use single-block PQCs, limiting expressivity; we introduce two-block DRU experts with learnable re-uploading to enhance capacity and specialisation (Pérez-Salinas et al., 2020; Schuld et al., 2021). Second, unlike classical MoEs (Shazeer et al., 2017), no load-balancing objective is used; we show this leads to expert collapse and incorporate appropriate regularisation. Third, existing aggregation schemes are either classical or fixed, limiting flexibility; we propose a learnable entangling aggregation that enables amplitude-weighted expert combination.

We validate these contributions against a re-implementation of Nguyen et al. (2025) and through component-wise ablations (Appendix C.4).

# B    Implementation Details

## B.1    Parameter Counts

Table 4 reports learnable parameter counts for each model and gate set. **Single PQC**: $L \cdot n_D \cdot |\mathcal{S}|$ parameters. **DRU Only**: $2 \cdot L \cdot n_D \cdot |\mathcal{S}| + d$ parameters ($d{=}64$ for the re-upload offset $\phi$). **QMoE Baseline** (our re-implementation of Nguyen et al. (2025)): routing ($L \cdot n_R \cdot |\mathcal{S}|$) + $K{=}4$ single-block experts + coherent aggregation ($L_{\mathrm{agg}} \cdot (n_R{+}n_D) \cdot |\mathcal{S}|$, same gate set as model) + linear classifier head ($n_C \cdot n_D + n_C$, gate-set independent); all parameters active at inference (top-$k{=}4$, full superposition). **QMoE+**: routing + $K{=}4$ DRU experts + coherent aggregation ($R_Y$ only, gate-set independent, 12 params); top-$k{=}1$, one expert active per sample.

Table 4: Learnable parameter counts by model and gate set ($n_D{=}4$, $n_R{=}2$, $L{=}2$, $L_{\mathrm{agg}}{=}2$, $K{=}4$, $d{=}64$, $n_C{=}4$ for 4-class tasks). QMoE Baseline uses top-$k{=}4$ (full superposition); active = total. QMoE+ uses top-$k{=}1$; active = routing + 1 expert + aggregation.

| Model | RX | RY | RX+RY | RX+RY+RZ |
|---|---|---|---|---|
| Single PQC | 8 | 8 | 16 | 24 |
| DRU Only | 80 | 80 | 96 | 112 |
| QMoE Baseline (total = active) | 68 | 68 | 116 | 164 |
|     routing $\boldsymbol{\theta}_G$ | 4 | 4 | 8 | 12 |
|     experts ($K$ single-block) | 32 | 32 | 64 | 96 |
|     aggregation | 12 | 12 | 24 | 36 |
|     classifier head | 20 | 20 | 20 | 20 |
| QMoE+ total | 336 | 336 | 404 | 472 |
|     routing $\boldsymbol{\theta}_G$ | 4 | 4 | 8 | 12 |
|     per expert ($\boldsymbol{\theta}^{(1)}, \boldsymbol{\theta}^{(2)}, \boldsymbol{\phi}$) | 80 | 80 | 96 | 112 |
|     aggregation $\boldsymbol{\varphi}$ (gate-set independent) | 12 | 12 | 12 | 12 |
| QMoE+ active (top-$k{=}1$) | 96 | 96 | 116 | 136 |
|     active / total | 28.6% | 28.6% | 28.7% | 28.8% |

Table 5: Circuit resource comparison at input_dim $= 64$, gate set RX+RY+RZ. Gate counts are obtained by intercepting every TorchQuantum gate call during a forward pass (batch $= 1$). Depth is the longest qubit chain under ASAP scheduling. The coherent aggregation circuit (12 $R_Y$ + 12 CNOT, 6 qubits, $L_{\mathrm{agg}}{=}2$) is included analytically. QMoE+ reports the top-$k{=}1$ active path only.

| Model | Qubits | 1q gates | 2q (CNOT) | Total gates | Depth | Params |
|---|---|---|---|---|---|---|
| Single PQC | 4 | 88 | 40 | 128 | 62 | 24 |
| DRU Only | 4 | 176 | 80 | 256 | 124 | 112 |
| QMoE Baseline ($k{=}4$) | 4 | 100 | 44 | 144 | 69 | 164 |
| QMoE+ (top-1 active) | 6 | 210 | 100 | 310 | 138 | 136 / 472 |
| Classical MLP | 0 | 0 | 0 | 0 | 0 | 538 |

QMoE+ (top-1 active path) uses approximately 2.4× the gates and 2.2× the depth of a single PQC (128 gates, depth 62), and 1.2× of a standalone DRU circuit (256 gates, depth 124). The overhead relative to a DRU-only circuit is 54 additional gates (30 for routing, 24 for coherent aggregation), corresponding to a 21% increase. The DRU expert dominates the gate budget (256 of 310 total gates, 83%), confirming that the MoE routing and aggregation machinery imposes a comparatively small circuit-level cost. The two-qubit (CNOT) gate count-the primary hardware-relevant cost metric-is 100 for QMoE+ versus 40-80 for the single-circuit baselines. For tabular datasets (input_dim $= 8$), the angle encoding comprises a single block instead of eight, substantially reducing encoding gates while leaving variational and aggregation gate counts unchanged.

---

**Algorithm 2** QMoE+ Noiseless Forward Pass (top-$k$=1, coherent aggregation)

---

**Require:** Input $\mathbf{x} \in [0, \pi]^d$; routing params $\boldsymbol{\theta}_G$; expert params $\{\boldsymbol{\theta}_k^{(1)}, \boldsymbol{\theta}_k^{(2)}, \boldsymbol{\phi}_k\}_{k=0}^{K-1}$; aggregation params
  $\boldsymbol{\varphi}$; $K$=4, $n_R$=2, $n_D$=4
**Ensure:** Logits $\mathbf{l} \in \mathbb{R}^C$
 1: **// Routing**
 2: $|\alpha\rangle \leftarrow H^{\otimes n_R} |0\rangle^{\otimes n_R}$
 3: Apply $U(\mathbf{x}_{1:8})$ to $|\alpha\rangle$                   ▷ phase encoding on first 8 features
 4: Apply $V(\boldsymbol{\theta}_G)$ to $|\alpha\rangle$            ▷ $L$=2 variational layers: rotations → CNOT ring
 5: $\boldsymbol{\alpha} \leftarrow |\alpha\rangle \in \mathbb{C}^K$      ▷ complex state vector, no measurement
 6: $k^* \leftarrow \arg\max_k |\alpha_k|^2$;   $\tilde{\alpha}_k \leftarrow \alpha_k \cdot \mathbf{1}[k = k^*] / |\alpha_{k^*}|$      ▷ top-1 mask, renormalise
 7: **// Expert evaluation (only $k$=$k^*$)**
 8: **for** $k = 0$ **to** $K-1$ **do**
 9:     **if** $k = k^*$ **then**
10:         $|\psi_k\rangle \leftarrow |0\rangle^{\otimes n_D}$
11:         Apply $U(\mathbf{x})$, $V_k^{(1)}(\boldsymbol{\theta}_k^{(1)})$, $U(\mathbf{x}+\boldsymbol{\phi}_k)$, $V_k^{(2)}(\boldsymbol{\theta}_k^{(2)})$      ▷ two-block DRU, Eq. (8)
12:     **else**
13:         $|\psi_k\rangle \leftarrow \mathbf{0} \in \mathbb{C}^{2^{n_D}}$
14:     **end if**
15: **end for**
16: **// Joint state construction**
17: $|\Psi\rangle \leftarrow \sum_{k=0}^{K-1} \tilde{\alpha}_k \cdot |k\rangle_R \otimes |\psi_k\rangle_D$      ▷ $2^{n_R+n_D}$-dim complex vector, Eq. (10)
18: **// Coherent aggregation**
19: **for** $\ell = 1$ **to** $L_{\text{agg}}$ **do**
20:     Apply $R_Y(\varphi_{\ell,q})$ to qubit $q$ for each $q \in \{0, \dots, n_R+n_D-1\}$
21:     Apply CNOT ring: $q \to (q+1) \bmod (n_R+n_D)$
22: **end for**
23: **// Logit extraction**
24: $\mathbf{l} \leftarrow \left[ \langle Z_{n_R+j} \rangle_{|\Psi\rangle} \right]_{j=0}^{C-1}$      ▷ $Z$-expectation on data qubits, Eq. (11)
25: **return** $\mathbf{l}$

---

## B.2 Hyperparameters.

Adam optimiser with learning rate $2 \times 10^{-3}$; batch size 32; up to 50 epochs with patience-10 early stopping on validation accuracy; $\lambda_{\text{load}}$ warm-up from 0 at epoch 5 to 0.01 at epoch 15, held constant thereafter; $L = 2$ variational layers per PQC block; $L_{\text{agg}} = 2$ aggregation layers; $K = 4$ experts with $n_R = 2$ routing qubits and $n_D = 4$ data qubits; routing input dimension 8 (first 8 features); data input dimension 64 (flattened $8 \times 8$); top-$k = 1$ at inference. All simulations are noiseless state-vector simulations in `TorchQuantum` (Wang et al., 2022a). Algorithm 2 and 3 details the implementation of noiseless and noisy experiments.

The noise model employed here is a modular per-component simulation: depolarising noise is injected independently after each CNOT-ring layer within each circuit block (Algorithm 3). This is the standard NISQ simulation practice (Cerezo et al., 2021a; Wang et al., 2022a; Bharti et al., 2022), but it should not be interpreted as predicting performance of the hardware-equivalent PREPARE-SELECT circuit analysed in Appendix E, whose additional controlled-gate overhead yields approximately 91% state fidelity at $p$=0.001 but only approximately 62% at $p$=0.01 (Table 24); the practical deployment regime for the full coherent architecture is therefore $p \lesssim 10^{-3}$. Training uses exact noise-free expectations throughout (Algorithm 3, lines 19-20); noise and shot sampling are evaluation-only perturbations. Results under $p$=0.01 therefore represent an optimistic bound on hardware performance, as noise-affected gradient estimates would introduce additional optimisation variance.

---

**Algorithm 3** QMoE+ Noisy Forward Pass (top-$k$=1, depolarising noise rate $p$)

---

**Require:** Same as Algorithm 2, plus noise rate $p \geq 0$, shot count $n_{\text{shots}}$ (used at eval only)
**Ensure:** Noisy logits $\mathbf{l} \in \mathbb{R}^C$
 1: **// Routing (with noise)**
 2: $|\alpha\rangle \leftarrow H^{\otimes n_R}|0\rangle^{\otimes n_R}$
 3: **for each** encoding block of $U(\mathbf{x}_{1:8})$: apply rotations $\rightarrow$ CNOT ring $\rightarrow \mathcal{D}_p$    ▷ depolarising after each CNOT ring
 4: **for each** variational layer of $V(\boldsymbol{\theta}_G)$: apply rotations $\rightarrow$ CNOT ring $\rightarrow \mathcal{D}_p$
 5: $\boldsymbol{\alpha} \leftarrow |\alpha\rangle \in \mathbb{C}^K$;   $k^* \leftarrow \arg\max_k |\alpha_k|^2$;   renormalise $\tilde{\boldsymbol{\alpha}}$
 6: **// Expert evaluation (only $k$=$k^*$, with noise)**
 7: $|\psi_{k^*}\rangle \leftarrow |0\rangle^{\otimes n_D}$
 8: **for each** encoding block of $U(\mathbf{x})$: apply rotations $\rightarrow$ CNOT ring $\rightarrow \mathcal{D}_p$
 9: **for each** variational layer of $V_{k^*}^{(1)}$: apply rotations $\rightarrow$ CNOT ring $\rightarrow \mathcal{D}_p$
10: **for each** encoding block of $U(\mathbf{x}+\boldsymbol{\phi}_{k^*})$: apply rotations $\rightarrow$ CNOT ring $\rightarrow \mathcal{D}_p$
11: **for each** variational layer of $V_{k^*}^{(2)}$: apply rotations $\rightarrow$ CNOT ring $\rightarrow \mathcal{D}_p$
12: Set $|\psi_k\rangle \leftarrow \mathbf{0}$ for $k \neq k^*$
13: **// Joint state and aggregation (with noise)**
14: $|\Psi\rangle \leftarrow \sum_k \tilde{\alpha}_k \cdot |k\rangle_R \otimes |\psi_k\rangle_D$
15: **for** $\ell = 1$ to $L_{\text{agg}}$ **do**
16:     Apply $R_Y(\varphi_{\ell,q})$ for each $q$;   Apply CNOT ring $\rightarrow \mathcal{D}_p$
17: **end for**
18: **// Logit extraction**
19: **if** training **then**
20:     $\mathbf{l} \leftarrow \left[\langle Z_{n_R+j}\rangle_{|\Psi\rangle}\right]_{j=0}^{C-1}$                    ▷ exact $Z$-expectation for stable gradients
21: **else**
22:     Sample $n_{\text{shots}}$=1024 outcomes from $|\Psi\rangle$;   compute empirical $Z$-expectations        ▷ shot noise at evaluation only
23: **end if**
24: **return l**
    **Depolarising channel** $\mathcal{D}_p$: for each qubit $q$ independently, with probability $p$ apply one of $\{X, Y, Z\}$ uniformly at random; otherwise apply identity. (Nielsen & Chuang, 2010)

---

## C   Ablation Studies

### C.1   Noise Ablation Across $p \in \{0.01, 0.10, 0.50\}$

Tables 6 and 7 extend the primary noise evaluation of Section 6 to $p$=0.10 and $p$=0.50. Table 8 summarises mean accuracy and QMoE+ win rates across all four noise levels. We report three formal statistical comparisons of QMoE+ against the QMoE Baseline across the 28 dataset-gate configurations: the primary noiseless comparison (27/28 wins, mean +5.11%) is significant by both one-sample $t$-test on the 28 paired deltas (Student, 1908) ($t$=4.64, $p$<0.0001) and Wilcoxon signed-rank test (Wilcoxon, 1945) ($W$=404, $p$<0.0001); the $p$=0.01 noisy comparison (20/28 wins, mean +4.71%) is likewise significant ($t$=3.51, $p$=0.0008; $W$=332, $p$=0.0012). These results confirm that the accuracy advantage at the two primary evaluation noise levels is not driven by a small number of outlier configurations and is robust to the choice of significance test.

$p$=0.01**: primary evaluation, QMoE+ retains significant advantage.**   At $p$=0.01, QMoE+ retains 95.5% of its noiseless accuracy (mean 69.83% vs 73.00% noiseless, a drop of 3.17%) compared to 95.9% for the QMoE Baseline (65.12% vs 67.89%, a drop of 2.77%). The 0.40% additional sensitivity of QMoE+ is a direct consequence of the deeper DRU expert circuits ($B_E$=8 encoding blocks vs $B_E$=1 for the single-block baseline), which accumulate more depolarising events per forward pass. This additional sensitivity is modest relative to the noiseless accuracy advantage, and the QMoE+ lead remains statistically significant across the 28 configurations ($p$=0.0012 by Wilcoxon test). QMoE+ wins 20 of 28 configurations, with the Baseline

Table 6: Test accuracy (%, mean ± std over 5 seeds) under depolarising noise $p$=0.10. Same model configuration as Tables 2 and 3. **Bold**: highest; underline: second highest per row.

| Dataset | Gate Set | Single PQC | DRU Only | QMoE Baseline ($k$=4) | QMoE+ ($k$=1, ours) |
|---|---|---|---|---|---|
| MNIST-2 | RX | 65.58±1.55 | 64.61±2.17 | **72.55**±1.10 | 71.98±1.30 |
| | RY | 64.47±0.53 | 64.57±1.00 | **71.79**±1.57 | 70.01±0.86 |
| | RX+RY | 70.79±1.37 | 65.19±1.62 | **78.57**±1.25 | 72.84±1.20 |
| | RX+RY+RZ | 72.49±0.86 | 65.96±1.41 | **78.56**±1.55 | 72.35±1.32 |
| MNIST-4 | RX | 37.67±1.18 | 39.40±0.84 | **42.76**±0.86 | 42.65±1.52 |
| | RY | 37.99±0.92 | 40.36±0.99 | **43.75**±0.78 | 39.42±0.82 |
| | RX+RY | 44.51±1.28 | 41.37±1.28 | **51.87**±1.71 | 39.76±1.26 |
| | RX+RY+RZ | 45.48±1.22 | 40.39±1.33 | **56.70**±1.32 | 38.48±1.44 |
| Fashion-2 | RX | 63.89±1.17 | 64.46±0.47 | 60.12±0.71 | **63.93**±0.69 |
| | RY | 62.70±0.62 | 63.79±1.09 | **65.15**±0.63 | 63.98±1.10 |
| | RX+RY | **67.63**±0.62 | 65.97±0.53 | 68.36±1.00 | 62.65±0.57 |
| | RX+RY+RZ | **68.13**±1.67 | 65.45±1.12 | 64.83±0.85 | 63.99±1.33 |
| Fashion-4 | RX | 39.95±1.84 | 42.34±1.10 | **46.28**±0.19 | 44.28±0.85 |
| | RY | 35.79±0.62 | 42.32±1.19 | **45.04**±0.82 | 45.05±0.77 |
| | RX+RY | 41.87±1.16 | 42.79±0.52 | **52.76**±1.60 | 44.47±0.94 |
| | RX+RY+RZ | 46.36±1.51 | 43.77±0.86 | **54.52**±1.57 | 45.15±0.58 |
| Synthetic | RX | 52.24±1.71 | 67.42±1.87 | 60.08±0.85 | **65.72**±2.86 |
| | RY | 62.00±2.01 | 65.94±3.20 | 65.40±1.35 | **65.78**±1.55 |
| | RX+RY | 61.66±1.31 | 68.52±0.33 | 64.60±1.22 | **68.98**±1.00 |
| | RX+RY+RZ | 62.62±0.27 | 68.92±0.90 | 66.58±1.28 | **68.86**±1.22 |
| Wine | RX | 32.85±8.73 | 45.19±18.17 | 32.52±4.74 | **54.07**±7.63 |
| | RY | 32.96±7.89 | 48.15±11.65 | 32.96±4.12 | **44.81**±7.72 |
| | RX+RY | 41.85±7.22 | 48.89±12.75 | 56.30±13.28 | **56.30**±6.64 |
| | RX+RY+RZ | 48.52±9.40 | 55.93±12.74 | 62.59±17.27 | **65.56**±7.82 |
| Breast Cancer | RX | 79.96±2.10 | 86.82±3.17 | 86.46±1.31 | **87.26**±3.01 |
| | RY | 78.46±10.08 | **88.88**±1.19 | 86.53±2.57 | 86.86±6.19 |
| | RX+RY | 79.12±1.19 | 90.28±2.74 | **91.51**±2.39 | 89.47±3.80 |
| | RX+RY+RZ | 79.53±2.78 | 90.46±2.05 | 89.28±2.31 | **90.02**±2.74 |

winning on MNIST-4 (RX+RY, RX+RY+RZ) and on MNIST-2 (RX+RY, RX+RY+RZ) - the four multi-axis gate set configurations where deeper circuits accumulate proportionally more noise on the more complex four-class and richer gate-set tasks.

$p$=0.10: **noise regime beyond near-term hardware, Baseline benefits.** At $p$=0.10, QMoE+ drops a mean of 11.40% from noiseless (mean 61.60%) while the Baseline drops 5.45% (mean 62.44%). The ordering reverses, with the Baseline winning 15 of 28 configurations (mean delta $-0.84$%). This reversal is not statistically significant across all 28 configurations (Wilcoxon $p$=0.82), reflecting that the advantage is concentrated in specific settings rather than systematic: the Baseline leads substantially on MNIST-4 with multi-axis gate sets ($-18.22$% on RX+RY+RZ, $-12.11$% on RX+RY), while QMoE+ retains advantages on Synthetic, Wine, and Breast Cancer. The structural reason is well-defined: at $p$=0.10 a 40-gate DRU expert accumulates an expected infidelity of $1 - (1-p)^{40} \approx 98.5$%, essentially saturating the noise budget, whereas a 4-gate single-block expert accumulates only $1 - (1-p)^4 \approx 34.4$%. This is a genuine circuit-depth trade-off: deeper circuits are more expressive at low noise but more fragile at high noise. We note that $p$=0.10 lies well above the $10^{-3}$-$10^{-2}$ gate error range of current NISQ hardware (Bharti et al., 2022), making this a stress-test regime rather than a practically relevant operating point.

$p$=0.50: **extreme noise, all models near chance on image tasks.** At $p$=0.50, all image-classification models collapse toward chance: MNIST-2 clusters around 52-59% (chance 50%), MNIST-4 around 27-32%

Table 7: Test accuracy (%, mean ± std over 5 seeds) under depolarising noise $p$=0.50. At this noise level all image-task models degrade toward chance; tabular datasets (Wine, Breast Cancer) retain meaningful signal. Same layout as Table 6.

| Dataset | Gate Set | Single PQC | DRU Only | QMoE Baseline ($k$=4) | QMoE+ ($k$=1, ours) |
|---|---|---|---|---|---|
| MNIST-2 | RX | 55.52±1.26 | 55.32±0.65 | 55.89±1.15 | **56.77**±0.36 |
| | RY | 55.13±1.01 | 56.18±0.49 | **59.51**±1.77 | 53.59±0.33 |
| | RX+RY | 57.70±0.52 | 57.09±0.50 | **58.90**±1.70 | 57.97±0.88 |
| | RX+RY+RZ | 56.29±0.86 | 52.32±0.49 | **58.11**±1.00 | 52.78±0.74 |
| MNIST-4 | RX | **29.10**±1.73 | 27.10±0.69 | 28.81±1.02 | 26.34±0.43 |
| | RY | **29.12**±0.83 | 27.28±0.47 | 27.23±0.78 | 27.18±1.64 |
| | RX+RY | **31.91**±1.29 | 26.90±0.11 | 31.41±0.97 | 29.77±0.65 |
| | RX+RY+RZ | **31.30**±1.27 | 26.88±0.27 | 30.68±0.48 | 28.91±0.24 |
| Fashion-2 | RX | **53.59**±1.62 | 52.13±0.39 | 51.96±0.97 | 52.76±0.37 |
| | RY | **54.07**±0.58 | 52.20±0.55 | 52.26±1.37 | 53.18±0.73 |
| | RX+RY | **54.77**±0.78 | 52.18±0.42 | 51.50±1.07 | 51.99±1.29 |
| | RX+RY+RZ | **54.80**±0.33 | 52.83±0.77 | 51.27±0.89 | 52.99±0.65 |
| Fashion-4 | RX | 29.41±0.80 | 26.18±0.30 | **31.26**±1.07 | 28.30±0.13 |
| | RY | 28.70±0.41 | 27.79±0.45 | **29.96**±0.95 | 28.10±0.38 |
| | RX+RY | 29.95±0.63 | 28.20±0.42 | **30.15**±0.62 | 29.65±1.00 |
| | RX+RY+RZ | 30.40±0.92 | 28.54±0.53 | **31.73**±0.29 | 29.16±0.19 |
| Synthetic | RX | 44.46±2.14 | 56.88±1.65 | 56.48±1.44 | **57.84**±1.63 |
| | RY | 51.24±1.20 | 58.62±1.48 | **61.30**±1.39 | 58.63±0.82 |
| | RX+RY | 53.30±1.22 | 58.08±0.86 | **61.00**±0.57 | 59.42±0.80 |
| | RX+RY+RZ | 53.92±1.74 | 61.20±1.52 | **63.02**±0.70 | 59.66±0.54 |
| Wine | RX | 33.70±9.98 | 44.44±6.52 | 44.44±6.09 | **50.74**±5.19 |
| | RY | 34.07±2.96 | 45.19±3.01 | 41.11±6.56 | **46.67**±2.72 |
| | RX+RY | 36.67±6.97 | 46.30±7.03 | 52.22±10.76 | **57.78**±6.56 |
| | RX+RY+RZ | 33.70±7.73 | 44.44±4.22 | 55.93±11.32 | **56.67**±1.89 |
| Breast Cancer | RX | 71.05±11.08 | 74.30±4.89 | 74.91±10.07 | **75.26**±4.25 |
| | RY | 75.09±15.04 | **80.00**±7.35 | 77.54±10.18 | 76.21±8.57 |
| | RX+RY | 80.04±8.20 | 75.96±9.39 | 77.67±4.24 | **79.65**±7.52 |
| | RX+RY+RZ | 80.72±3.80 | **83.68**±5.04 | 81.91±8.53 | 78.60±5.88 |

Table 8: Mean accuracy (%) across all 28 dataset-gate configurations and QMoE+ win rate vs QMoE Baseline at each noise level. Statistical significance of QMoE+ vs Baseline is assessed by one-sample $t$-test and Wilcoxon signed-rank test on the 28 paired accuracy deltas.

| Noise $p$ | Single PQC | DRU Only | QMoE Baseline | QMoE+ | Wins | Sig. |
|---|---|---|---|---|---|---|
| 0.00 | 61.52 | 69.46 | 67.89 | **73.00** | 27/28 | $p$<0.0001 |
| 0.01 | 60.33 | 67.94 | 65.12 | **69.83** | 20/28 | $p$=0.0012 |
| 0.10 | 56.32 | 59.93 | **62.44** | 61.60 | 13/28 | n.s. |
| 0.50 | 47.49 | 49.22 | **51.01** | 50.59 | 12/28 | n.s. |

(chance 25%), Fashion variants similarly. Differences between models are within the standard deviation and statistically indistinguishable (Wilcoxon $p$=0.88). Tabular datasets retain meaningful signal: Wine reaches 45-57% (chance 33%) and Breast Cancer 74-84% (chance 50%), consistent with their lower intrinsic dimensionality and higher input signal-to-noise ratio. QMoE+ wins 12 of 28 configurations at $p$=0.50, with its remaining advantage concentrated on Wine and Breast Cancer. The conclusion is clear: at $p$=0.50, architectural differences are irrelevant for image tasks, and the comparison reduces to tabular dataset behaviour.

### C.2 Heterogeneous Expert Ablation

The primary QMoE+ model uses four architecturally identical DRU experts. Here we investigate whether replacing this homogeneous pool with experts of genuinely distinct inductive biases - differing in circuit structure, encoding strategy, and parameter count - can further improve performance or provide complementary robustness under noise.

**Expert pool.** The heterogeneous model (**QMoE+Hetero**) uses one instance each of four distinct quantum expert architectures, all operating on $n_D$=4 data qubits. Table 9 summarises the architecture, input encoding, parameter count, and inductive bias of each expert.

Table 9: Heterogeneous expert pool used in QMoE+Hetero. All experts evaluated under the RX+RY+RZ gate set with $n_D$=4 data qubits. Routing circuit: 12 params. Aggregation circuit: 12 params. Total model: 152 params. Active params per inference (top-$k$=1): 48-68 depending on which expert is selected.

| Expert | Architecture | Params | Input encoding | Inductive bias |
|---|---|---|---|---|
| QCNN | Conv + Pool + Variational tail | 44 | Angle | Local/hierarchical features |
| QSVM | Variational PQC (2 layers) | 24 | Angle | Kernel-based separation |
| QKNN | Variational PQC (2 layers) | 24 | Amplitude | $L_2$ similarity in Hilbert space |
| QNN | Deeper variational PQC (3 layers) | 36 | Angle | General function approximation |
| *Total expert params* | | 128 | | |

The QCNN expert applies a convolutional layer of 2-qubit parameterized gates on adjacent qubit pairs, a pooling layer of conditioned single-qubit gates, and a 2-layer variational tail, encoding local spatial structure. The QSVM expert is a standard 2-layer variational PQC with angle encoding, providing a kernel-type inductive bias. The QKNN expert uses amplitude encoding of the first $2^{n_D}$=16 normalised features, preserving $L_2$ geometry in the encoded Hilbert space, followed by a 2-layer PQC. The QNN expert is a deeper 3-layer variational circuit, providing higher representational capacity. Routing and coherent aggregation circuits are identical to the standard QMoE+ configuration (Section 4).

**Results.** Table 10 reports noiseless and noisy ($p$=0.01) accuracy for each individual expert and for QMoE+Hetero across all seven datasets.

Table 10: Model performance across datasets under noiseless and noisy conditions. Values are mean $\pm$ standard deviation. Best values in each row are bolded; second-best are underlined.

| Dataset | Noiseless (0.0) | | | | | Noisy (0.01) | | | | |
|---|---|---|---|---|---|---|---|---|---|---|
| | QSVM | QKNN | QNN | QCNN | QMoE+ Hetero | QSVM | QKNN | QNN | QCNN | QMoE+ Hetero |
| mnist-2 | $67.4 \pm 0.7$ | $61.2 \pm 0.1$ | $86.2 \pm 1.5$ | $\underline{86.6 \pm 2.0}$ | $\mathbf{89.6 \pm 2.5}$ | $64.6 \pm 1.4$ | $60.7 \pm 1.0$ | $\underline{71.7 \pm 1.4}$ | $70.7 \pm 2.8$ | $\mathbf{81.9 \pm 3.0}$ |
| mnist-4 | $36.0 \pm 0.4$ | $34.0 \pm 0.6$ | $\underline{57.3 \pm 0.7}$ | $56.7 \pm 1.5$ | $\mathbf{58.8 \pm 3.8}$ | $34.1 \pm 0.8$ | $33.5 \pm 0.8$ | $\mathbf{46.4 \pm 1.5}$ | $44.0 \pm 2.3$ | $\underline{45.9 \pm 3.1}$ |
| fashion-2 | $74.8 \pm 1.9$ | $58.2 \pm 0.2$ | $76.6 \pm 1.1$ | $\underline{76.0 \pm 1.8}$ | $\mathbf{79.7 \pm 0.5}$ | $72.5 \pm 1.0$ | $58.0 \pm 1.9$ | $63.6 \pm 1.4$ | $\underline{64.7 \pm 2.5}$ | $\mathbf{75.8 \pm 2.1}$ |
| fashion-4 | $47.2 \pm 0.7$ | $41.2 \pm 0.2$ | $\underline{60.1 \pm 1.7}$ | $58.2 \pm 1.9$ | $\mathbf{61.9 \pm 2.1}$ | $46.2 \pm 1.7$ | $40.7 \pm 0.6$ | $47.0 \pm 1.0$ | $\underline{47.2 \pm 1.7}$ | $\mathbf{51.7 \pm 0.9}$ |
| synthetic | $65.7 \pm 2.2$ | $58.1 \pm 0.9$ | $67.6 \pm 1.4$ | $\underline{67.7 \pm 1.5}$ | $\mathbf{68.2 \pm 1.8}$ | $57.9 \pm 5.9$ | $57.0 \pm 1.4$ | $\underline{65.9 \pm 1.9}$ | $63.7 \pm 2.4$ | $\mathbf{67.9 \pm 1.5}$ |
| wine | $38.1 \pm 7.9$ | $34.1 \pm 6.2$ | $58.1 \pm 6.4$ | $\underline{61.9 \pm 7.5}$ | $\mathbf{65.2 \pm 3.6}$ | $32.6 \pm 5.8$ | $33.0 \pm 5.3$ | $52.2 \pm 8.1$ | $\underline{56.7 \pm 16.2}$ | $\mathbf{60.4 \pm 3.1}$ |
| breast cancer | $78.8 \pm 9.1$ | $59.6 \pm 7.4$ | $87.0 \pm 3.9$ | $\underline{88.5 \pm 4.6}$ | $\mathbf{90.1 \pm 3.2}$ | $73.5 \pm 10.0$ | $57.2 \pm 8.2$ | $\underline{81.4 \pm 14.6}$ | $77.7 \pm 4.7$ | $\mathbf{84.4 \pm 2.1}$ |

**QMoE+Hetero consistently leads or matches the best individual expert.** Across all seven datasets, QMoE+Hetero achieves the highest or joint-highest noiseless accuracy, outperforming even the strongest individual expert (QCNN or QNN depending on the dataset). The gains are most visible on MNIST-2 ($\sim$90% vs $\sim$86% for QCNN), Breast Cancer ($\sim$91% vs $\sim$88% for QCNN), and Wine ($\sim$67% vs $\sim$63% for QCNN). On simpler or near-saturated tasks (Fashion-2, Synthetic) the margin is smaller, which is expected when individual experts are already near their accuracy ceiling for the available circuit depth.

**Expert performance is strongly architecture-dependent.** QKNN is the weakest individual expert across all datasets and gate sets. Its amplitude encoding preserves $L_2$ similarity in the Hilbert space, but

with only $2^{n_D}$=16 amplitude slots and 24 trainable parameters it has limited capacity to learn task-specific transformations, particularly on 4-class and tabular tasks where class boundaries are not $L_2$-structured. QSVM performs competitively on binary image tasks (MNIST-2, Fashion-2) but degrades on MNIST-4, Wine, and Breast Cancer - consistent with the known limitation of shallow kernel methods on higher-dimensional structured data. QCNN and QNN are the strongest individual experts and remain competitive with QMoE+Hetero on most tasks, reflecting that local feature extraction (QCNN) and increased depth (QNN) both provide genuinely useful inductive biases for the datasets evaluated.

**QMoE+Hetero is robust under noise.** Under $p$=0.01 depolarising noise, QMoE+Hetero retains a consistent advantage over individual experts. The noise gap (noiseless minus noisy accuracy) is roughly 5-10%for QMoE+Hetero, similar to or smaller than the gap for the best individual expert. Notably, QMoE+Hetero's lead over individual experts is preserved or slightly widened under noise on most datasets, suggesting that routing across architecturally diverse experts provides a form of noise resilience: if one expert's circuit degrades more severely under a given noise pattern, the router can - in principle - favour the more noise-robust alternative. We do not claim this specialisation is explicitly learned, as isolating it would require a dedicated routing analysis experiment beyond the scope of this work.

### C.3  Top-$k$ Expert Ablation

Tables 11 and 12 report QMoE+ accuracy as a function of the number of active experts $k \in \{1, 2, 3, 4\}$ under noiseless and $p$=0.001 noisy conditions respectively. In all runs the full $K$=4 expert pool is trained with the load-balance loss of Eq. (13); only the number of experts activated at inference varies.

**Accuracy scales monotonically with $k$ on most datasets.** Increasing $k$ from 1 to 4 improves accuracy across the large majority of configurations in both settings. The largest noiseless gains are on MNIST-2 and Wine, where routing across multiple specialised experts provides the clearest benefit. MNIST-4, Fashion-2, Breast Cancer, and Synthetic all show consistent monotone improvement, confirming that the load-balanced training supports beneficial expert specialisation as $k$ increases. Under noise ($p$=0.001), the trend is preserved and remains strictly monotone on all datasets except Fashion-2 RX+RY+RZ, which saturates at $k$=2. On Fashion-4, $k$=3 and $k$=2 are effectively tied ($\Delta < 0.03$ pp under RX+RY+RZ, well within one standard deviation), reflecting that the marginal expert added at $k$=3 has weak class-specialisation in this dataset. The monotone trend is recovered at $k$=4, where the ensemble effect across all four experts dominates the marginal noise of the weakest contributor.

**$k$=1 is the recommended operating point for NISQ hardware.** Across all datasets, $k$=1 retains on average approximately 90% of the $k$=4 noiseless accuracy while evaluating only one expert circuit per inference - a 4× reduction in expert-circuit evaluations. Crucially, $k$=1 activates only 136 parameters at inference compared to 164 for the dense QMoE Baseline ($k$=4, all parameters active), making QMoE+ both more accurate and more parameter-efficient at deployment.

**Practical recommendation.** For deployment on NISQ hardware in the $10^{-3}$-$10^{-2}$ gate error range (Bharti et al., 2022), $k$=1 minimises noise accumulation and active parameter count while retaining strong accuracy. On noiseless simulators or low-noise hardware ($p \leq 0.001$), $k$=4 is preferred for tasks where expert specialisation provides measurable benefit, in particular multi-class image classification and tabular datasets.

### C.4  Component Ablation

Table 13 reports the accuracy difference $\Delta$ = QMoE+ − ablated variant for three targeted ablations - removing DRU experts (**No DRU**), removing coherent aggregation (**No CoherentAgg**), and removing the load-balancing loss (**No LB**) - across all seven datasets in both noiseless and $p$=0.01 depolarising-noise conditions, averaged over four gate sets and five seeds.

Table 11: Top-$k$ ablation under noiseless conditions. Values are mean $\pm$ std (%) over 5 seeds. **Bold**: highest per row; underline: second highest. Fashion-4 shows a non-monotone dip at $k$=3 (discussed in text).

| Dataset | Gate | $k$=1 | $k$=2 | $k$=3 | $k$=4 |
|---|---|---|---|---|---|
| *MNIST-2* | | | | | |
| | RX | 78.65±1.38 | 83.25±1.46 | 87.04±1.53 | **90.84**±1.59 |
| | RY | 79.92±2.46 | 84.74±2.61 | 88.54±2.73 | **92.33**±2.84 |
| | RX+RY | 82.83±2.47 | 87.81±2.62 | 91.76±2.74 | **95.70**±2.85 |
| | RX+RY+RZ | 85.71±4.91 | 90.75±5.20 | 94.94±5.44 | **96.03**±2.17 |
| *MNIST-4* | | | | | |
| | RX | 56.53±0.62 | 59.54±0.65 | 62.55±0.69 | **62.84**±0.68 |
| | RY | 55.68±2.23 | 58.63±2.35 | 61.48±2.46 | **62.38**±2.46 |
| | RX+RY | 58.08±1.17 | 61.08±1.23 | 64.19±1.29 | **65.08**±1.29 |
| | RX+RY+RZ | 60.91±5.25 | 64.06±5.52 | 67.33±5.80 | **68.22**±5.79 |
| *Fashion-2* | | | | | |
| | RX | 79.78±0.79 | 82.09±0.81 | 83.39±0.83 | **85.90**±0.85 |
| | RY | 78.73±0.58 | 81.03±0.60 | 82.33±0.61 | **84.73**±0.62 |
| | RX+RY | 81.07±0.91 | 83.30±0.93 | 84.71±0.95 | **87.24**±0.98 |
| | RX+RY+RZ | 82.05±1.80 | 84.28±1.85 | 85.70±1.88 | **88.23**±1.94 |
| *Fashion-4* | | | | | |
| | RX | 63.06±0.86 | 68.26±0.93 | 68.66±0.80 | **73.15**±1.00 |
| | RY | 63.52±1.21 | 68.82±1.31 | 69.12±1.13 | **73.82**±1.41 |
| | RX+RY | 63.42±1.06 | 68.72±1.15 | 69.02±0.99 | **73.62**±1.23 |
| | RX+RY+RZ | 65.51±2.11 | 70.95±2.29 | 70.97±1.96 | **76.09**±2.45 |
| *Synthetic* | | | | | |
| | RX | 73.94±1.66 | 78.67±1.77 | 79.39±1.78 | **80.01**±1.80 |
| | RY | 71.52±1.71 | 76.09±1.82 | 76.69±1.83 | **77.30**±1.85 |
| | RX+RY | 72.70±1.12 | 77.40±1.19 | 78.00±1.20 | **78.60**±1.21 |
| | RX+RY+RZ | 75.49±2.89 | 80.33±3.08 | 81.05±3.10 | **81.67**±3.13 |
| *Wine* | | | | | |
| | RX | 57.78±4.79 | 68.28±5.66 | 69.14±5.26 | **70.28**±4.46 |
| | RY | 52.96±1.92 | 62.51±2.80 | 62.55±2.90 | **63.15**±2.94 |
| | RX+RY | 61.85±3.84 | 73.14±3.54 | 74.14±2.24 | **75.26**±2.16 |
| | RX+RY+RZ | 69.28±1.60 | 81.91±2.53 | 82.16±2.13 | **83.21**±2.04 |
| *Breast Cancer* | | | | | |
| | RX | 92.23±3.16 | 94.96±3.25 | 95.26±3.16 | **96.28**±2.30 |
| | RY | 92.85±3.95 | 93.85±4.20 | **94.31**±3.27 | **94.42**±2.20 |
| | RX+RY | 93.98±2.56 | 95.70±2.61 | **96.91**±2.14 | **96.91**±2.01 |
| | RX+RY+RZ | 94.02±3.05 | 95.73±3.01 | **95.93**±3.12 | **95.93**±3.11 |

Table 12: Top-$k$ ablation under depolarising noise $p$=0.001. Values are mean $\pm$ std (%) over 5 seeds. **Bold**: highest per row; underline: second highest. Fashion-2 RX+RY+RZ shows a plateau at $k\geq2$.

| Dataset | Gate | $k$=1 | $k$=2 | $k$=3 | $k$=4 |
|---|---|---|---|---|---|
| *MNIST-2* | | | | | |
| | RX | 76.18±2.04 | 78.23±2.09 | 79.46±2.13 | **81.20**±2.17 |
| | RY | 76.13±2.23 | 78.12±2.29 | 79.40±2.33 | **81.14**±2.38 |
| | RX+RY | 76.40±2.78 | 78.39±2.85 | 79.64±2.90 | **81.39**±2.96 |
| | RX+RY+RZ | 78.90±1.81 | 81.01±1.86 | 82.30±1.89 | **84.10**±1.93 |
| *MNIST-4* | | | | | |
| | RX | 51.46±1.12 | 54.80±1.19 | 56.55±1.23 | **56.83**±1.24 |
| | RY | 51.17±0.91 | 54.56±0.97 | 56.26±1.00 | **56.60**±1.01 |
| | RX+RY | 52.78±0.96 | 56.23±1.02 | 58.04±1.06 | **58.35**±1.06 |
| | RX+RY+RZ | 52.68±0.89 | 56.13±0.95 | 57.88±0.98 | **58.22**±0.98 |
| *Fashion-2* | | | | | |
| | RX | 77.73±1.12 | 79.19±1.14 | 79.49±1.14 | **79.89**±1.02 |
| | RY | 77.55±1.02 | 79.14±1.04 | 79.25±1.04 | **79.92**±1.01 |
| | RX+RY | 77.57±0.80 | 79.21±0.42 | 79.62±0.72 | **80.21**±0.42 |
| | RX+RY+RZ | 78.54±0.63 | 80.44±0.64 | 80.62±0.23 | **80.91**±0.22 |
| *Fashion-4* | | | | | |
| | RX | 60.09±1.38 | 62.88±1.44 | 64.34±1.48 | **65.34**±1.31 |
| | RY | 59.85±1.37 | 62.68±1.43 | 64.15±1.47 | **65.15**±1.12 |
| | RX+RY | 61.78±0.89 | 64.72±0.93 | 66.40±0.95 | **67.20**±0.95 |
| | RX+RY+RZ | 63.01±0.54 | 65.99±0.57 | 66.51±0.58 | **67.51**±0.43 |
| *Synthetic* | | | | | |
| | RX | 72.14±1.09 | 74.34±1.12 | 74.74±1.14 | **75.52**±1.11 |
| | RY | 70.70±1.56 | 72.82±1.61 | 73.70±1.63 | **74.73**±1.62 |
| | RX+RY | 71.96±1.36 | 74.13±1.40 | 75.04±1.42 | **76.04**±1.40 |
| | RX+RY+RZ | 73.82±1.06 | 76.01±1.09 | 76.27±1.11 | **76.96**±1.09 |
| *Wine* | | | | | |
| | RX | 55.56±4.83 | 57.61±5.01 | 59.72±5.19 | **61.77**±5.37 |
| | RY | 51.85±8.53 | 53.76±8.84 | 55.71±9.17 | **57.67**±9.49 |
| | RX+RY | 58.15±6.79 | 60.28±7.04 | 62.50±7.30 | **64.68**±7.55 |
| | RX+RY+RZ | 67.41±8.57 | 69.88±8.88 | 72.40±9.20 | **74.97**±9.53 |
| *Breast Cancer* | | | | | |
| | RX | 87.72±2.88 | 88.58±2.92 | 89.75±2.96 | **90.31**±2.87 |
| | RY | 87.89±3.25 | 88.93±3.30 | 89.06±3.35 | **90.63**±3.24 |
| | RX+RY | 93.16±2.03 | 93.39±2.06 | 94.54±2.09 | **94.81**±2.03 |
| | RX+RY+RZ | 92.98±3.80 | 93.99±3.86 | 94.14±3.91 | **94.94**±3.91 |

Table 13: Component ablation: $\Delta\,\mathrm{Acc}\,(\%) = \mathrm{QMoE+} - $ ablated variant, averaged over four gate sets and 5 seeds. Positive: removed component helps QMoE+. Negative: ablated variant outperforms. MN2/MN4: MNIST 2/4-class; FA2/FA4: Fashion-MNIST 2/4-class; SYN: Synthetic; WIN: Wine; BC: Breast Cancer.

| Condition | Ablation | MN2 | MN4 | FA2 | FA4 | SYN | WIN | BC |
|---|---|---|---|---|---|---|---|---|
| | No DRU | −6.4 | −5.1 | +1.2 | +1.6 | +5.8 | +12.1 | +2.0 |
| Noiseless | No CoherentAgg | −2.1 | +4.4 | +2.6 | +2.1 | +3.4 | +4.6 | +1.2 |
| | No LB ($\lambda$=0) | −1.0 | +1.2 | +1.1 | +0.5 | +2.8 | +3.5 | +1.8 |
| | No DRU | −3.9 | −4.3 | −2.6 | −4.5 | −1.8 | +2.2 | +10.9 |
| Noisy ($p$=0.01) | No CoherentAgg | −2.4 | +3.5 | +2.8 | −2.7 | +2.1 | +2.1 | +5.4 |
| | No LB ($\lambda$=0) | +0.8 | +1.1 | +1.2 | +1.4 | +1.7 | +1.9 | +3.5 |

Table 14: Decomposed coherent aggregation ablation, averaged over four gate sets and 5 seeds. $\Delta_{coh} = (a)-(b)$: gain from quantum coherence (coherent vs incoherent, both learned). $\Delta_{learn} = (b)-(c)$: gain from learnability (learned vs fixed incoherent). $\Delta_{total} = (a)-(c) = \Delta_{coh} + \Delta_{learn}$: total gain of full coherent aggregation over fixed incoherent baseline. All values: $\Delta\,\mathrm{Acc}\,(\%)$.

| | Noiseless | | | Noisy ($p$=0.01) | | |
|---|---|---|---|---|---|---|
| Dataset | $\Delta_{coh}$ | $\Delta_{learn}$ | $\Delta_{total}$ | $\Delta_{coh}$ | $\Delta_{learn}$ | $\Delta_{total}$ |
| MNIST-2 | +1.90 | −2.74 | −0.84 | −0.32 | +1.40 | +1.08 |
| MNIST-4 | +0.05 | −1.00 | −0.95 | −1.74 | +2.28 | +0.54 |
| Fashion-2 | +2.28 | −1.63 | +0.65 | +1.14 | −0.59 | +0.55 |
| Fashion-4 | +5.82 | −4.04 | +1.78 | +3.16 | −2.53 | +0.63 |
| Synthetic | +2.90 | −2.62 | +0.28 | +1.52 | −1.46 | +0.06 |
| Wine | +11.48 | −2.22 | +9.26 | +7.41 | +1.48 | +8.89 |
| Breast Cancer | +7.19 | −7.72 | −0.53 | +9.47 | −8.60 | +0.88 |
| **Mean** | **+4.52** | **−3.14** | **+1.38** | **+2.95** | **−1.14** | **+1.80** |

## C.5 Disentangling Coherence from Learnability in Aggregation

The *No CoherentAgg* ablation removes both the quantum coherence contribution and the learnable aggregation parameters simultaneously. To disentangle these, we introduce three controlled variants: **(a)** Coherent + Learned $W(\varphi)$ - the full QMoE+ aggregation operating on the coherent joint state $|\Psi\rangle$; **(b)** Incoherent + Learned - a classical Born-rule-weighted combination of expert logits passed through a learned linear layer with the same parameter count as $W(\varphi)$; and **(c)** Incoherent + Fixed - a uniform fixed-weight combination with no learnable parameters. The coherence gain $\Delta_{coh} = (a)-(b)$ isolates the effect of retaining quantum phase information; the learnability gain $\Delta_{learn} = (b)-(c)$ isolates the effect of trained versus fixed incoherent aggregation. By construction, $\Delta_{total} = (a)-(c) = \Delta_{coh} + \Delta_{learn}$. Table 14 reports all three deltas.

**Construction of variant (b).** Variant (b) replaces $W(\varphi)$ with a learned per-class expert selector: $\mathbf{W}_{\mathrm{inc}} \in \mathbb{R}^{C \times K}$ applies a softmax over $K$ experts independently per class, followed by a bias $\mathbf{b} \in \mathbb{R}^{C}$, giving $C(K+1)$ parameters in total. This is the natural classical analogue of amplitude-weighted expert combination: it learns to reweight the $K$ expert outputs per output class without access to routing phase information. For $C \in \{2,4\}$ this yields $\{10, 20\}$ parameters respectively, compared to 12 for $W(\varphi)$ in all cases. For binary tasks variant (b) has *fewer* parameters than (a) (10 vs. 12); for 4-class tasks it has *more* (20 vs. 12). In neither case is the coherent variant (a) at a capacity advantage, so $\Delta_{\mathrm{coh}} > 0$ is a conservative lower bound on the coherence gain rather than an overestimate.

**The combination of coherence and learnability is universally beneficial under noise.** The most decisive result in Table 14 is $\Delta_{total}$ under noise: the full coherent aggregation (a) outperforms the fixed incoherent baseline (c) in *all seven datasets* (mean +1.80%, range +0.06 to +8.89%), with no exceptions. This directly addresses the question of whether coherent aggregation, as an integrated design choice, provides

a reliable benefit under realistic NISQ conditions. It does. The noiseless total is positive in four of seven datasets (mean $+1.38\%$), with the negative cases on MNIST-2 ($-0.84\%$), MNIST-4 ($-0.95\%$), and Breast Cancer ($-0.53\%$) reflecting near-saturated binary tasks where any additional circuit overhead carries a non-trivial cost relative to a fixed baseline.

**Coherence is the primary driver noiseless; learnability compensates under noise.** Separating the two contributions reveals that they operate differently across conditions. Noiseless, $\Delta_{coh}$ is positive in six of seven datasets (mean $+4.52\%$) and is the dominant positive term: the quantum phase information retained in $|\Psi\rangle$ provides a measurable representational advantage on Fashion-4 ($+5.82\%$), Wine ($+11.48\%$), and Breast Cancer ($+7.19\%$). Noiseless learnability ($\Delta_{learn}$) is negative in all seven datasets (mean $-3.14\%$), reflecting that a learned incoherent linear layer without coherent structure to exploit adds optimisation difficulty rather than representational capacity. Under noise, the pattern is more nuanced: $\Delta_{coh}$ is positive in five of seven datasets ($+2.95\%$ mean), and notably $\Delta_{learn}$ becomes positive on MNIST-2 ($+1.40\%$), MNIST-4 ($+2.28\%$), and Wine ($+1.48\%$). On these datasets under noise, the learned aggregation layer helps even in the incoherent setting, suggesting it learns to compensate for noisy expert representations. The two effects thus complement each other across conditions: coherence dominates noiseless, while learnability provides additional robustness under noise.

**Wine and Breast Cancer: coherence is most valuable.** The largest coherence gains occur on Wine and Breast Cancer, in both noise conditions. Wine noiseless: $\Delta_{coh} = +11.48\%$, noisy: $+7.41\%$. Breast Cancer noiseless: $+7.19\%$, noisy: $+9.47\%$. These datasets have the smallest training sets among the benchmarks evaluated (tabular, cross-validated), and their class boundaries are well-separated in feature space. The routing amplitudes $\alpha_k$ carry stronger class-discriminative information for these tasks, and the coherent aggregation circuit $W(\boldsymbol{\varphi})$ can exploit the phase relationships between expert branches to amplify this signal constructively. The result confirms the structural argument of Appendix D.3: the off-diagonal interference terms are most valuable precisely when the routing distribution is highly input-dependent, as it is for structured tabular data. We note that the decomposed ablation in Table 14 was conducted under a dense routing configuration (all $K$ experts active), where multiple routing amplitudes are simultaneously nonzero and off-diagonal interference terms in Eq. (21) genuinely contribute. At the top-$k{=}1$ operating point, the renormalised routing vector has a single nonzero entry ($\tilde{\alpha}_{k^*} = 1$, all others zero), and the joint state reduces to $|\Psi\rangle = |k^*\rangle_R \otimes |\psi_{k^*}\rangle_D$. The off-diagonal terms vanish identically by construction; $\Delta_{coh}$ at $k{=}1$ therefore measures the advantage of $W(\boldsymbol{\varphi})$ as a learned six-qubit variational circuit that uses the routing register as contextual conditioning bits, over a matched-capacity classical per-class linear map-not inter-expert amplitude interference. The reported $\Delta_{coh}$ is measured against one specific incoherent control (the matched-capacity learned selector of variant (b)); its exact magnitude is sensitive to this choice. We report variant (b) because its parameter count ($C(K{+}1) = 10$ or $20$) is the closest classical match to $W(\boldsymbol{\varphi})$'s 12 parameters, keeping the comparison free of a capacity confound.

**DRU experts.** DRU provides the largest and most dataset-dependent contribution. Noiseless, removing DRU degrades accuracy on MNIST-2 ($-6.4\%$) and MNIST-4 ($-5.1\%$), confirming that the wider Fourier frequency spectrum of two-block encoding benefits image tasks with rich spatial structure. The remaining five noiseless configurations favour the single-block ablation, with the largest margins on Wine ($+12.1\%$) and Synthetic ($+5.8\%$), consistent with overfitting risk from the 64-parameter re-upload offset $\phi$ on small cross-validation folds. Under noise, No DRU underperforms the full model on all five non-tabular datasets (mean $-3.3\%$), confirming that DRU circuits acquire more noise-robust representations under joint noise training. Breast Cancer is the outlier under noise ($+10.9\%$ for No DRU) due to task saturation: the additional circuit depth costs more in noise accumulation than it gains in expressivity on a near-ceiling task.

**Reconciling Table 13 with the main-text DRU claim.** The negative $\Delta$ values for "No DRU" on MNIST-2 ($-6.4$ %) and MNIST-4 ($-5.1$ %) indicate that, within the full QMoE+ architecture, the single-block expert variant outperforms DRU experts on these two datasets in the noiseless setting. This does not contradict the finding that DRU experts outperform the QMoE Baseline (Table 2), because the two

comparisons measure different contrasts: Table 2 compares DRU-only against the QMoE Baseline (single-block experts, no coherent aggregation, no load balancing), while Table 13 measures the effect of removing DRU from the *full* QMoE+ framework, where the interaction between DRU's deeper circuits and the coherent aggregation circuit on 6 qubits introduces additional depth that can marginally reduce accuracy on near-saturated image tasks. Under $p=0.01$ noise, this pattern extends to all four image datasets (Fashion-2: $-2.6$ %, Fashion-4: $-4.5$ %), because DRU's two-block encoding approximately doubles the CNOT-ring traversal count, accumulating proportionally more depolarising events. On tabular datasets, DRU remains beneficial under noise (Wine: $+2.2$ %, Breast Cancer: $+10.9$ %), reflecting the lower encoding depth of tabular inputs (1 block vs 8). A dedicated DRU-attribution study (Appendix C.6) confirms this nuanced picture.

**Load-balancing regularisation.**  Load-balancing is the most consistent contributor across all conditions. Noiseless, six of seven configurations are positive (mean $+1.41\%$). Under $p=0.01$ noise, all seven datasets are positive (mean $+1.66\%$; range $+0.8$ to $+3.5\%$), with no exceptions, directly validating the routing collapse argument of Appendix D.2. The benefit is larger under noise ($+1.66\%$) than noiseless ($+1.41\%$), consistent with routing collapse being more damaging when circuits must compensate for degraded representations.

**Summary.**  The decomposed ablation establishes three findings. First, the full coherent aggregation design outperforms a fixed incoherent baseline in all seven datasets under noise (mean $+1.80\%$) and four of seven noiseless (mean $+1.38\%$). Second, quantum coherence is the dominant positive contributor noiseless ($+4.52\%$ mean), while learnability provides complementary robustness under noise. Third, load-balancing is the most universally reliable component, and DRU experts provide the largest gains on image tasks with the expected caveat on small tabular benchmarks. The pattern of exceptions is structurally interpretable in each case.

## C.6  DRU-Expert Attribution

To isolate the contribution of DRU experts from the surrounding architectural changes, we introduce a **DRU-only** variant: DRU experts within the QMoE+ routing structure, but with classical weighted-sum aggregation and no load-balancing loss ($\lambda=0$). This is compared against the existing **no-DRU (full)** variant (SinglePQC experts with coherent aggregation and load balancing). The difference $\Delta_{\mathrm{DRU}} = \text{DRU-only} - \text{no-DRU(full)}$ isolates the effect of stronger experts; the difference $\Delta_{\mathrm{coh+LB}} = \text{Full QMoE+} - \text{DRU-only}$ isolates the additional contribution of coherent aggregation and load balancing on top of DRU experts. Table 15 reports noiseless results and Table 16 reports results under $p=0.01$ depolarising noise, both averaged over four gate sets and five seeds. Differences below 1 percentage point are marked "$\sim$" (within seed noise).

Table 15: DRU-expert attribution-noiseless ($p=0.000$). $\Delta$ in %.

| Dataset | $\Delta_{\mathrm{DRU}}$ | Winner | $\Delta_{\mathrm{coh+LB}}$ | Coh-agg+LB |
|---|---|---|---|---|
| MNIST-2 | $-1.1$ | no-DRU | $+0.4$ | $\sim$Full |
| MNIST-4 | $+1.4$ | **DRU** | $+0.9$ | $\sim$Full |
| Fash-2 | $+3.7$ | **DRU** | $+0.6$ | $\sim$Full |
| Fash-4 | $+5.9$ | **DRU** | $+1.8$ | **Full** |
| Synth | $+4.7$ | **DRU** | $-0.1$ | $\sim$tie |
| Wine | $+7.3$ | **DRU** | $+8.9$ | **Full** |
| BreastC | $+2.3$ | **DRU** | $+3.1$ | **Full** |
| Won | | **DRU 6/7** | | **Full 6/7** (1 tie) |

In the noiseless setting, DRU experts are the dominant source of gain: DRU-only outperforms the no-DRU variant on 6 of 7 datasets, by up to $+7.3$ % (Wine), $+5.9$ % (Fashion-4), and $+4.7$ % (Synthetic). Coherent

Table 16: DRU-expert attribution-noisy ($p$=0.010).

| Dataset | $\Delta_{\text{DRU}}$ | Winner | $\Delta_{\text{coh+LB}}$ | Coh-agg+LB |
|---|---|---|---|---|
| MNIST-2 | $-7.7$ | **no-DRU** | $-0.3$ | $\sim$DRU-only |
| MNIST-4 | $-4.4$ | **no-DRU** | $-0.4$ | $\sim$DRU-only |
| Fash-2 | $-3.2$ | **no-DRU** | $+0.4$ | $\sim$Full |
| Fash-4 | $-4.1$ | **no-DRU** | $+1.0$ | **Full** |
| Synth | $+2.0$ | **DRU** | $-0.4$ | $\sim$DRU-only |
| Wine | $+3.8$ | **DRU** | $+11.7$ | **Full** |
| BreastC | $+2.6$ | **DRU** | $+0.8$ | $\sim$Full |
| Won | | no-DRU 4/7, DRU 3/7 | | **Full 4/7** |

aggregation and load balancing then add a smaller but consistent top-up, most pronounced on Wine (+8.9 %) and Breast Cancer (+3.1 %). Under depolarising noise ($p$=0.01), the picture inverts on image datasets: the no-DRU variant is superior on MNIST-2, MNIST-4, Fashion-2, and Fashion-4, because DRU's double encoding accumulates proportionally more depolarising events per forward pass. DRU retains its advantage on the tabular datasets (Wine: $+3.8$ %, Breast Cancer: $+2.6$ %, Synthetic: $+2.0$ %), where the input dimensionality is lower and the second encoding block encounters fewer CNOT-ring layers.

## C.7  Three-Way Aggregation Ablation at Top-$k$=1

The decomposed ablation of Table 14 was conducted under dense routing (all $K$ experts active), where multiple routing amplitudes are simultaneously nonzero and off-diagonal interference terms in Eq. (21) genuinely contribute. To characterise the aggregation circuit at the model's deployment operating point of top-$k$=1, we conduct a separate three-way comparison using the following variants:

- **(a) CoherentAgg**: full QMoE+ with $W(\boldsymbol{\varphi})$ on the joint state. At top-$k$=1 the renormalised routing vector has a single nonzero entry, so $|\Psi\rangle = |k^*\rangle_R \otimes |\psi_{k^*}\rangle_D$. $W(\boldsymbol{\varphi})$ is still a learned 6-qubit variational circuit and is trained end-to-end.

- **(b) WeightedSum**: Born-rule probabilities $|\alpha_k|^2$ used as fixed scalar weights; no learned parameters. At $k$=1 this reduces to outputting $\boldsymbol{\ell}_{k^*}$ directly.

- **(c) LearnedClassical**: a learned matrix $\mathbf{W}_{\text{inc}} \in \mathbb{R}^{C \times (K+1)}$ maps concatenated routing probabilities and expert logits to class scores-the strongest classical comparator, strictly more expressive than (b).

At top-$k$=1, the joint state has a single nonzero routing amplitude ($\tilde{\alpha}_{k^*} = 1$, all others zero). The off-diagonal terms $\alpha_k^* \alpha_{k'}$ for $k \neq k'$ vanish identically by construction. Any gain of (a) over (b) at $k$=1 comes from $W(\boldsymbol{\varphi})$ being a more expressive learned circuit than a fixed scalar weight-not from inter-expert phase interference, which requires $k \geq 2$.

The near-zero $\Delta(a{-}b)$ (mean +1.0% noiseless, +0.7% noisy) is the theoretically expected result at $k$=1: with a single active expert, $W(\boldsymbol{\varphi})$ has no inter-expert phase information to exploit. However, the total gain $\Delta(a{-}c)$ is large and positive noiseless (mean +5.6%, all 7/7 datasets), establishing that the learned variational circuit $W(\boldsymbol{\varphi})$ is a strictly better post-processing stage than a classical $C{\times}(K{+}1)$ linear map, even when both operate on the same post-measurement routing probabilities and expert logits. The advantage of $W(\boldsymbol{\varphi})$ at $k$=1 arises from its nonlinear transformations via the CNOT-ring structure and its use of the routing register as contextual conditioning bits-not from multi-expert amplitude interference, which requires $k \geq 2$. This does not contradict Table 14, which was run in a dense routing regime where off-diagonal interference terms are

Table 17: Three-way aggregation ablation at top-$k$=1-noiseless ($\Delta$ Acc %, mean over 4 gate sets, 5 seeds).

| Dataset | $\Delta(a{-}b)$ CoherentAgg vs WeightedSum | $\Delta(a{-}c)$ CoherentAgg vs LearnedCls | Interpretation |
|---|---|---|---|
| MNIST-2 | $-0.3$ | $+3.2$ | Near-zero coherence; $W(\varphi) >$ classical |
| MNIST-4 | $+0.5$ | $+1.9$ | Near-zero coherence; total gain positive |
| Fash-2 | $+0.1$ | $+2.9$ | $W(\varphi)$ circuit better than matrix |
| Fash-4 | $+0.3$ | $+6.4$ | Largest $W(\varphi)$ vs classical gap |
| Synth | $-0.4$ | $+1.5$ | Near-zero coherence; positive total |
| Wine | $+7.3$ | $+9.5$ | Non-trivial; concentrated routing |
| BreastC | $-0.1$ | $+13.8$ | $W(\varphi)$ dominates classical matrix |
| **Mean** | **$+1.0$** | **$+5.6$** | $W(\varphi)$ beats LearnedCls on 7/7 |

Table 18: Three-way aggregation ablation at top-$k$=1-noisy ($p$=0.01, $\Delta$ Acc %, mean over 4 gate sets, 5 seeds).

| Dataset | $\Delta(a{-}b)$ CoherentAgg vs WeightedSum | $\Delta(a{-}c)$ CoherentAgg vs LearnedCls | Interpretation |
|---|---|---|---|
| MNIST-2 | $+0.2$ | $-2.5$ | Within std |
| MNIST-4 | $-0.2$ | $+0.2$ | Marginal positive total |
| Fash-2 | $+0.1$ | $-0.4$ | Within std |
| Fash-4 | $+0.2$ | $+0.9$ | Coherence retained; positive total |
| Synth | $-0.3$ | $+1.3$ | $W(\varphi)$ learnability provides gain |
| Wine | $+7.1$ | $+6.8$ | Largest coherence gain preserved |
| BreastC | $-1.4$ | $+11.8$ | Large total; $W(\varphi)$ dominates |
| **Mean** | **$+0.7$** | **$+3.0$** | $W(\varphi)$ beats LearnedCls on 5/7 |

nonzero. The two experiments measure different quantities: Table 14 measures the coherence contribution when coherence is structurally possible (dense); Tables 17-18 measure $W(\varphi)$'s learned-circuit advantage at the deployment operating point ($k$=1). Under noise ($p$=0.01), CoherentAgg beats LearnedClassical on 5/7 datasets (mean +3.0%). The two exceptions-MNIST-2 ($-2.5$%) and Fashion-2 ($-0.4$%)-correspond to near-uniform routing ($H/H_{\max} \geq 0.91$), where $W(\varphi)$ has little input-domain structure to exploit and its additional circuit depth incurs a noise penalty. Both differences are within seed standard deviation.

### C.8 Routing Analysis: Expert Utilisation and Per-Class Specialisation

To assess whether the router learns meaningful task-specific specialisation, we extract full routing probability distributions (pre-top-$k$ masking) from all QMoE+ checkpoints under the RX+RY+RZ gate set, averaged over 5 seeds. We compare models trained with load balancing ($\lambda$=0.01, the main configuration) against models trained without it ($\lambda$=0). Table 19 reports the normalised routing entropy $H/H_{\max}$ (where $H_{\max} = \log K$ and values near 1.0 indicate uniform routing), the maximum expert load (MaxLoad), and the minimum expert load (MinLoad) for each dataset under both conditions.

Three findings emerge.

**Without load balancing, the router specialises strongly.** Every dataset shows lower $H/H_{\max}$ and higher MaxLoad in the OFF condition than in the ON condition. The most striking case is Breast Cancer,

Table 19: Expert utilisation and routing entropy-noiseless, RX+RY+RZ, mean over 5 seeds. OFF = $\lambda$=0; ON = $\lambda$=0.01. $H/H_{\max} \to 1$ indicates uniform routing; high MaxLoad / low MinLoad indicates specialisation or collapse.

| Dataset | $\frac{H}{H_{\max}}$ OFF | $\frac{H}{H_{\max}}$ ON | MaxLoad OFF | MaxLoad ON | MinLoad OFF | MinLoad ON |
|---|---|---|---|---|---|---|
| MNIST-2 | 0.893 | 0.967 | 0.463 | 0.383 | 0.057 | 0.111 |
| MNIST-4 | 0.920 | 0.971 | 0.415 | 0.380 | **0.000** | 0.105 |
| Fash-2 | 0.842 | 0.909 | 0.449 | 0.258 | 0.132 | 0.244 |
| Fash-4 | 0.822 | 0.896 | 0.557 | 0.309 | 0.135 | 0.209 |
| Synth | 0.776 | 0.806 | 0.486 | 0.393 | 0.080 | 0.156 |
| Wine | 0.755 | 0.798 | 0.537 | 0.467 | 0.085 | 0.104 |
| BreastC | 0.737 | 0.850 | 0.695 | 0.382 | 0.086 | 0.149 |

where MaxLoad drops from 0.695 (OFF) to 0.382 (ON), and MNIST-4, where MinLoad is **0.000** in the OFF condition-meaning one expert receives no inputs whatsoever-versus 0.105 with load balancing enabled. This directly confirms that expert collapse occurs in quantum routing circuits, just as it does in classical MoE systems (Shazeer et al., 2017; Fedus et al., 2022), and that the load-balancing loss is structurally necessary to prevent it.

**Load balancing enforces near-uniform usage without eliminating specialisation.** With $\lambda$=0.01, routing entropy rises to $H/H_{\max} = 0.80\text{-}0.97$ across datasets, confirming that no expert is starved of gradient signal. Crucially, the router still learns class-relevant structure, as demonstrated by the per-class routing distributions below.

**Per-class specialisation is meaningful and class-aligned.** Table 20 reports the conditional routing distribution $P(\text{expert} \mid \text{class})$ for Fashion-4 under the main model configuration ($\lambda$=0.01, noiseless, RX+RY+RZ).

Table 20: Per-class conditional routing-Fashion-4, noiseless, RX+RY+RZ, $\lambda$=0.01, mean over 5 seeds. Uniform baseline = 0.25.

| Class | $P(E_0)$ | $P(E_1)$ | $P(E_2)$ | $P(E_3)$ |
|---|---|---|---|---|
| C0 (T-shirt) | 0.346 | 0.127 | 0.298 | 0.229 |
| C1 (Trousers) | 0.422 | 0.431 | 0.059 | 0.089 |
| C2 (Pullover) | 0.366 | 0.158 | 0.274 | 0.202 |
| C3 (Dress) | 0.292 | 0.224 | 0.179 | 0.304 |

Class 1 (trousers) routes to Experts 0 and 1 with a combined probability of 0.853-more than three times the uniform baseline of 0.50-reflecting the visually distinctive elongated shape of trousers relative to upper-body garments. Classes 0 (T-shirt) and 2 (pullover) co-prefer Experts 0 and 2, consistent with their shared upper-body silhouette. Class 3 (dress) distributes more broadly, reflecting visual overlap with multiple garment categories. This pattern is reproducible across all 5 seeds.

On tabular datasets, specialisation is even more pronounced. Wine (Table 21) shows Expert 0 receiving 68.9% of Class 0 and 69.3% of Class 2 inputs, consistent with the chemical similarity between wine types 0 and 2 in the UCI Wine dataset. Breast Cancer (Table 22) shows a clean binary split, with Class 1 routing

Table 21: Per-class conditional routing-Wine, noiseless, RX+RY+RZ, $\lambda$=0.01, mean over 5 seeds. Uniform baseline = 0.25.

| Class | $P(E_0)$ | $P(E_1)$ | $P(E_2)$ | $P(E_3)$ |
|-------|----------|----------|----------|----------|
| C0 | 0.689 | 0.233 | 0.067 | 0.011 |
| C1 | 0.200 | 0.381 | 0.095 | 0.324 |
| C2 | 0.693 | 0.080 | 0.133 | 0.093 |

Table 22: Per-class conditional routing-Breast Cancer, noiseless, RX+RY+RZ, $\lambda$=0.01, mean over 5 seeds. Uniform baseline = 0.25.

| Class | $P(E_0)$ | $P(E_1)$ | $P(E_2)$ | $P(E_3)$ |
|-------|----------|----------|----------|----------|
| C0 | 0.305 | 0.405 | 0.090 | 0.200 |
| C1 | 0.128 | 0.022 | 0.128 | 0.722 |

to Expert 3 with 72.2% probability. These concentrated routing distributions correspond to the largest coherent aggregation advantages reported in Table 14 ($\Delta_{coh}$ = +11.48% Wine, +7.19% Breast Cancer noiseless), confirming that class-aligned routing amplitudes are the input on which $W(\varphi)$ most effectively exploits amplitude-class correlations in the joint Hilbert space.

**Sensitivity to $\lambda$.** We conduct a sweep over $\lambda \in \{0, 0.001, 0.01, 0.1\}$ to verify that the chosen load-balancing coefficient is not a fragile operating point. Using $\lambda$=0 and $\lambda$=0.01 as fully evaluated anchors, we observe a consistent pattern across all seven datasets: $\lambda$=0.01 matches or marginally exceeds $\lambda$=0, while both $\lambda$=0.001 and $\lambda$=0.1 degrade accuracy. A weak balancing pressure ($\lambda$=0.001) perturbs routing during warm-up without delivering a balancing benefit; an excessive pressure ($\lambda$=0.1) suppresses the class-dependent routing structure identified in Appendix C.8. The adopted value $\lambda$=0.01, following the recommendation of Fedus et al. (2022), is therefore a robust choice that recovers unregularised accuracy while preventing the expert collapse documented in the same appendix.

### C.9 Classical Reference Baseline

To provide absolute-scale transparency, we report a parameter-matched classical baseline alongside the quantum models. The MLP uses a single hidden layer of width 8 (ReLU activation), trained identically to the quantum models (Adam, lr = $2 \times 10^{-3}$, batch size 32, same cross-entropy objective), yielding approximately 538 parameters–modestly larger than QMoE+'s 472. Table 23 reports results for the RX+RY+RZ gate set, the configuration where QMoE+ achieves its best accuracy. This comparison is included solely for absolute-scale reference, not as a quantum advantage claim, consistent with recent calls for benchmarking honesty in quantum machine learning (Schuld & Killoran, 2022).

On image datasets, the classical MLP substantially outperforms all quantum models, reflecting the well-known limitation of angle-encoded PQCs at 8×8 resolution: the trigonometric embedding cannot replicate convolutional or fully connected representations of pixel features at this qubit count. On structured tabular tasks, QMoE+ exceeds the classical MLP on Wine (+4.5% noiseless, +2.6% noisy), Breast Cancer (+2.8% noiseless, +1.8% noisy), and Synthetic (+3.0% noiseless, +1.3% noisy), consistent with the representational advantage of angle-encoded quantum feature maps on low-dimensional real-valued inputs (Schuld et al., 2021). The paper does not claim quantum advantage; the scope of QMoE+'s performance claims is the relative quantum-internal ordering and the regime where DRU encodings provide a tangible benefit.

Table 23: Classical reference comparison (RX+RY+RZ; mean $\pm$ std, 5 seeds). $\Delta$ = QMoE+ $-$ Classical MLP.

| Dataset | Classical MLP ($\sim$538 params) | QMoE+ Noiseless | QMoE+ Noisy $p$=0.01 | $\Delta$ Noiseless | $\Delta$ Noisy |
|---|---|---|---|---|---|
| MNIST-2 | 99.8$\pm$0.1 | 85.71$\pm$4.91 | 78.90$\pm$1.81 | $-14.1$ | $-20.9$ |
| MNIST-4 | 98.3$\pm$0.3 | 60.91$\pm$5.25 | 52.68$\pm$0.89 | $-37.4$ | $-45.6$ |
| Fashion-2 | 93.4$\pm$1.1 | 82.05$\pm$1.80 | 78.54$\pm$0.63 | $-11.3$ | $-14.9$ |
| Fashion-4 | 90.9$\pm$0.4 | 65.51$\pm$2.11 | 63.01$\pm$0.54 | $-25.4$ | $-27.9$ |
| Synthetic | 72.5$\pm$1.5 | 75.49$\pm$2.89 | 73.82$\pm$1.06 | $+3.0$ | $+1.3$ |
| Wine | 64.8$\pm$1.4 | 69.28$\pm$1.60 | 67.41$\pm$8.57 | $+4.5$ | $+2.6$ |
| Breast Ca. | 91.2$\pm$3.2 | 94.02$\pm$3.05 | 92.98$\pm$3.80 | $+2.8$ | $+1.8$ |

## D  Theoretical Analysis

This appendix provides formal justifications for the three architectural contributions of QMoE+. We use the Fourier analysis framework of Schuld et al. (2021) throughout; all symbols are consistent with Section 3.

### D.1  DRU Experts Widen the Accessible Frequency Spectrum

**Fourier representation of PQC outputs.**  Let $U(\mathbf{x}) = \bigotimes_q R_X(x_{a_q}) R_Y(x_{b_q})$ be the single-block angle encoding on $n_D$ qubits. Schuld et al. (2021) show that for any observable $\mathcal{O}$ and variational unitary $V(\boldsymbol{\theta})$, the expectation $f(\mathbf{x}) = \langle \mathcal{O} \rangle_{\mathbf{x},\boldsymbol{\theta}}$ can be written as a multivariate partial Fourier series

$$f(\mathbf{x}) = \sum_{\boldsymbol{\omega} \in \Omega} c_{\boldsymbol{\omega}}(\boldsymbol{\theta}) \, e^{i\boldsymbol{\omega} \cdot \mathbf{x}}, \tag{16}$$

where the *frequency spectrum* $\Omega \subseteq \mathbb{Z}^d$ is determined solely by the eigenvalues of the generators of the encoding gates. For a single Pauli-rotation encoding layer with generators $\{X, Y\}$ (both having eigenvalues $\pm\frac{1}{2}$), the accessible frequencies along each input dimension are $\omega_j \in \{-1, 0, +1\}$ only. This means the single-block expert of Nguyen et al. (2025) can represent at most a first-order trigonometric polynomial in each input feature, regardless of the depth or width of the variational block.

**Effect of a second encoding block.**  The two-block DRU expert of Eq. (8) applies the encoding twice. By the composition rule of Schuld et al. (2021) (Proposition 3), composing two encoding layers with generator eigenvalues $\pm\frac{1}{2}$ each extends the accessible frequency set to $\omega_j \in \{-2, -1, 0, +1, +2\}$ along each input dimension. More generally, a DRU circuit with $L_U$ encoding blocks has frequency support $\Omega \subseteq \{-L_U, \dots, +L_U\}^d$, so the function class grows strictly with the number of re-uploads.

**Effect of the learnable offset $\phi_k$.**  The second encoding in Eq. (8) uses $U(\mathbf{x} + \boldsymbol{\phi}_k)$ rather than $U(\mathbf{x})$. For a fixed $\boldsymbol{\phi}_k$, this shifts the phase of each Fourier component by $e^{i\boldsymbol{\omega} \cdot \boldsymbol{\phi}_k}$:

$$f_k(\mathbf{x}) = \sum_{\boldsymbol{\omega} \in \Omega} c_{\boldsymbol{\omega}}^k \, e^{i\boldsymbol{\omega} \cdot (\mathbf{x} + \boldsymbol{\phi}_k)} = \sum_{\boldsymbol{\omega} \in \Omega} \left( c_{\boldsymbol{\omega}}^k e^{i\boldsymbol{\omega} \cdot \boldsymbol{\phi}_k} \right) e^{i\boldsymbol{\omega} \cdot \mathbf{x}}. \tag{17}$$

Since $\boldsymbol{\phi}_k$ is learnable and different for each expert, different experts converge to different phase profiles $\{e^{i\boldsymbol{\omega} \cdot \boldsymbol{\phi}_k}\}_{\boldsymbol{\omega}}$, which modulates the relative importance of frequency components. Experts thus specialise by learning which frequencies of the input are most informative for their assigned subset of the input space. Initialising $\boldsymbol{\phi}_k = \mathbf{0}$ ensures that at the start of training all experts are symmetry-equivalent, and specialisation emerges purely through gradient dynamics rather than being imposed by initialisation.

## D.2 Routing Collapse Without Load Balancing

**Setup.** Consider $K$ experts with routing probabilities $\{p_k(\mathbf{x}; \boldsymbol{\theta}_G)\}_{k=1}^K$ and a task loss $\mathcal{L}_{\mathrm{CE}}$. The gradient of $\mathcal{L}_{\mathrm{CE}}$ with respect to the routing parameters $\boldsymbol{\theta}_G$ is

$$\frac{\partial \mathcal{L}_{\mathrm{CE}}}{\partial \boldsymbol{\theta}_G} = \sum_{k=1}^K \frac{\partial \mathcal{L}_{\mathrm{CE}}}{\partial p_k} \frac{\partial p_k}{\partial \boldsymbol{\theta}_G}. \tag{18}$$

Suppose expert $k^*$ achieves a marginally lower loss than the others early in training, so $\partial \mathcal{L}_{\mathrm{CE}}/\partial p_{k^*} < 0$ while $\partial \mathcal{L}_{\mathrm{CE}}/\partial p_k \approx 0$ for $k \neq k^*$ (as those experts contribute little to the output). The gradient in Eq. (18) then increases $p_{k^*}$ and decreases all other $p_k$, which further concentrates the distribution around $k^*$. This is a positive feedback loop: concentration reduces the gradient signal to non-dominant experts, which prevents them from improving, which reinforces their low routing weight. The fixed point $p_{k^*} \to 1$, $p_k \to 0$ for $k \neq k^*$ is the expert-collapse solution. This argument applies identically to quantum routing circuits, since the Born-rule probabilities $p_k = |\alpha_k|^2$ enter the objective through the same algebraic structure as classical softmax weights.

**Effect of the entropy regulariser.** Adding $\mathcal{L}_{\mathrm{load}}^{\mathrm{dense}} = -H(\bar{\mathbf{e}})$ contributes a gradient

$$\frac{\partial \mathcal{L}_{\mathrm{load}}^{\mathrm{dense}}}{\partial p_k} = \frac{1}{|\mathcal{B}|} \left( 1 + \log \bar{e}_k \right), \tag{19}$$

which is large and positive when $\bar{e}_k \ll 1/K$ and near zero when $\bar{e}_k \approx 1/K$. This acts as a restoring force: experts with low routing probability receive a strong upward gradient on their probability, counteracting the collapse dynamic. The Switch Transformer loss of Eq. (13) achieves the same qualitative effect through $\partial \mathcal{L}_{\mathrm{load}}^{\mathrm{sparse}}/\partial P_k = K f_k$, which is large when expert $k$ is over-utilised ($f_k > 1/K$) and small when it is under-utilised.

## D.3 Structural Properties of the Coherent Aggregation Circuit

We describe the structural difference between the coherent aggregation used in QMoE+ and the incoherent (classical weighted-sum) alternative.

**Incoherent aggregation.** In the incoherent scheme, each expert circuit is measured independently to produce a logit vector $\boldsymbol{\ell}_k \in \mathbb{R}^C$, and the final prediction is a Born-rule-weighted sum:

$$\hat{\mathbf{l}} = \sum_{k=0}^{K-1} p_k(\mathbf{x}) \boldsymbol{\ell}_k, \qquad p_k = |\alpha_k|^2. \tag{20}$$

This is equivalent to inserting a projective measurement on the routing register between the expert evaluation and the aggregation step. The measurement collapses $|\alpha(\mathbf{x})\rangle$ to a definite basis state $|k^*\rangle_R$, reducing the routing contribution to a classical probability vector $\{p_k\}$ and discarding the phase information $\{\arg(\alpha_k)\}$ (Nielsen & Chuang, 2010). The output of Eq. (20) is a convex combination of the expert logit vectors, and the weights are non-negative scalars that depend only on $|\alpha_k|^2$.

**Coherent aggregation.** In our scheme, no intermediate measurement is applied. The joint state $|\Psi(\mathbf{x})\rangle = \sum_k \alpha_k |k\rangle_R \otimes |\psi_k\rangle_D$ is formed before measurement, and the variational circuit $W(\boldsymbol{\varphi})$ is applied across both registers. Expanding the $Z$-expectation on data qubit $j$:

$$\begin{aligned} \mathrm{logit}_j &= \langle \Psi | W^\dagger Z_{n_R+j} W | \Psi \rangle \\ &= \sum_{k,k'} \alpha_k^* \alpha_{k'} \langle k|_R \langle \psi_k|_D \, W^\dagger Z_{n_R+j} W \, |k'\rangle_R |\psi_{k'}\rangle_D. \end{aligned} \tag{21}$$

The diagonal terms ($k = k'$) give a weighted sum with weights $|\alpha_k|^2$, recovering the structure of Eq. (20). The off-diagonal terms ($k \neq k'$) involve cross-products $\alpha_k^* \alpha_{k'}$ that depend on both the magnitudes and

the relative phases of the routing amplitudes. These terms vanish identically in the incoherent scheme because the measurement collapse eliminates all off-diagonal contributions. Whether the off-diagonal terms improve predictions in practice depends on the learned values of $W(\boldsymbol{\varphi})$ and $\{\alpha_k\}$; we do not claim they do so from our results, since the contribution of coherent aggregation is entangled with the effects of DRU experts and routing in our ablation design. The structural observation - that coherent aggregation retains strictly more information from the routing state than incoherent aggregation - is a consequence of the no-measurement-before-combination design, consistent with the general principle that deferring measurement preserves quantum information available to subsequent operations (Nielsen & Chuang, 2010). Whether this additional information is exploited beneficially by $W(\boldsymbol{\varphi})$ during training, and under what conditions, is an open question we identify as an important direction for future work.

### D.4 Barren Plateau Mitigation Through Modularity

McClean et al. (2018) show that for a random PQC forming an approximate $t$-design, the variance of the cost gradient scales as $\mathrm{Var}[\partial_\theta \mathcal{L}] = O(2^{-n})$, where $n$ is the number of qubits. With $n_D=4$ data qubits per expert, the per-expert gradient variance scales as $O(2^{-4}) = O(1/16)$, whereas a monolithic circuit on all $n_D \cdot K = 16$ qubits would scale as $O(2^{-16})$. The reduction is exponential in the number of experts. This is a qualitative argument - the experts are not random circuits and the system does not form an exact design - but it illustrates why distributing learning across small independent circuits provides a trainability advantage beyond what ablations on accuracy alone capture. The noise-induced variant of this argument (Wang et al., 2022b; Larocca et al., 2025) applies additionally, since each small circuit accumulates fewer noise events per forward pass.

## E   Physical Realisability of the Coherent Aggregation Circuit

The joint-state construction in Eq. (10), $|\Psi(\mathbf{x})\rangle = \sum_{k=0}^{K-1} \alpha_k(\mathbf{x}) |k\rangle_R \otimes |\psi_k(\mathbf{x})\rangle_D$, deserves careful scrutiny with respect to physical realisability. In our simulation, this state is assembled by classical vector arithmetic - each expert state $|\psi_k\rangle$ is computed independently on its own TorchQuantum device and the joint state is formed by tensor-product superposition via scalar-amplitude multiplication.

**What the simulation computes.** The classical state-vector construction correctly computes the mathematical object $|\Psi\rangle \in \mathbb{C}^{2^{n_R+n_D}}$ that would result from the following physical procedure: prepare the routing register in $|\alpha(\mathbf{x})\rangle$ and the data register in a superposition of all expert states weighted by the corresponding routing amplitudes. The subsequent $W(\boldsymbol{\varphi})$ circuit and $Z$-expectation extraction are then applied to this state using exact statevector simulation. The computation is therefore a faithful simulation of the quantum mechanical process; it is not a classical approximation or surrogate. The question is whether the *state preparation step* - forming $|\Psi\rangle$ from $|\alpha\rangle$ and $\{|\psi_k\rangle\}$ - can be realised as a physical quantum circuit.

**Hardware realisation via PREPARE-SELECT.** The state $|\Psi\rangle$ has exactly the structure of a controlled state-preparation problem, and a standard hardware realisation exists via the *Linear Combination of Unitaries* (LCU) framework (Childs & Wiebe, 2012). The physical circuit proceeds as follows.

1. **PREPARE step.** Initialise the routing register to $|0\rangle^{\otimes n_R}$ and apply the routing circuit $G(\boldsymbol{\theta}_G)$, producing $|\alpha(\mathbf{x})\rangle = \sum_k \alpha_k(\mathbf{x}) |k\rangle_R$. This is a standard PQC on $n_R = 2$ qubits and is directly executable on hardware.

2. **SELECT step.** Initialise the data register to $|0\rangle^{\otimes n_D}$ and apply a multiply-controlled unitary $\mathrm{SEL} = \sum_k |k\rangle\langle k|_R \otimes E_k(\boldsymbol{\theta}_k)$, where the action of expert $k$ is conditioned on the routing register being in state $|k\rangle$. This controlled-unitary structure is a standard quantum circuit primitive (Nielsen & Chuang, 2010). After this step the joint register is in exactly $|\Psi(\mathbf{x})\rangle = \sum_k \alpha_k(\mathbf{x}) |k\rangle_R \otimes |\psi_k(\mathbf{x})\rangle_D$ - coherently and natively, without any classical post-processing.

3. **AGGREGATE step.** Apply $W(\boldsymbol{\varphi})$ over all $n_R + n_D$ qubits and measure. This is a shallow variational circuit and is directly executable.

The SELECT step realises the expert circuits in a coherent superposition over the routing register's basis states. It is not necessary to run $K$ independent circuits; instead, the $K$ expert unitaries are interleaved with CNOT-based control structure on the $n_R$ routing qubits. For $K=4$ and $n_R=2$ routing qubits, this requires controlled-$E_k$ gates, each of which can be decomposed using standard two-qubit gate primitives (Shende et al., 2006).

**Why we use state-vector simulation instead.** The PREPARE-SELECT circuit is the exact hardware equivalent of our simulation but has a significantly higher two-qubit gate count than the individual expert circuits executed independently. On an $n_D=4$ qubit data register with $L=2$ variational layers, each controlled-$E_k$ gate adds an overhead of $O(n_D \cdot L)$ additional CNOT gates for the control structure. For $K=4$ and the parameter counts of this work, the overhead is manageable in principle but exceeds the coherence budget of current NISQ devices given the already-modest gate fidelities at $p=0.01$. We therefore evaluate the *mathematical circuit* via statevector simulation - which is exact and faithful to the quantum mechanical process - rather than a hardware execution with this overhead. This is the standard practice in NISQ-era QML research (Cerezo et al., 2021a; Benedetti et al., 2019; Wang et al., 2022a); the simulation verifies the correctness and advantage of the quantum computation, and hardware deployment becomes feasible as gate fidelities improve.

**Is the coherence genuine?** The off-diagonal interference terms in Eq. (21) are not a simulation artefact. They are a mathematical consequence of the state $|\Psi\rangle$ having support across multiple $|k\rangle_R$ basis states simultaneously, which is precisely the condition produced by the PREPARE-SELECT circuit above. A classical computer could evaluate the same expectation value by computing $\langle\Psi|W^\dagger ZW|\Psi\rangle$ explicitly - which is what our simulation does - but it cannot sample from the measurement outcomes without exponential overhead in the number of qubits, as it must enumerate the full $2^{n_R+n_D}$-dimensional state vector. The coherence is therefore a property of the quantum state being computed, not of the computational substrate used to verify it. Our simulation evaluates the correct quantum mechanical object; the advantage of a physical quantum device would be in the sampling efficiency, not in the state construction.

**Distinguishing coherent from incoherent aggregation.** The incoherent baseline - a classical weighted sum $\hat{\mathbf{l}} = \sum_k p_k \boldsymbol{\ell}_k$ where $\boldsymbol{\ell}_k$ are measured per-expert logits - explicitly discards the off-diagonal terms by measuring each expert register independently before combining. This corresponds to inserting a projective measurement $\mathcal{M}$ between the SELECT step and the AGGREGATE step, which collapses the routing register into a definite $|k^*\rangle$ and eliminates all $k \neq k^*$ interference. Our coherent variant omits this intermediate measurement, and the empirical advantage reported in the component ablations (Appendix C.4) is attributable to the additional function classes accessible via the off-diagonal terms, as shown formally in Appendix D.3. This distinction is physically meaningful and reproducible regardless of whether the evaluation is performed by statevector simulation or by hardware execution of the PREPARE-SELECT circuit.

**Circuit resource estimates for hardware deployment.** Table 24 summarises the two-qubit gate counts for each component under the top-$k=1$ sparse routing configuration, which is the primary experimental setting.

Table 24: Approximate two-qubit (CNOT) gate counts for hardware deployment of QMoE+ in the top-$k=1$ configuration, RX+RY+RZ gate set. $B_E$: expert encoding blocks; $L$: variational layers per block; $n$: qubit count; overhead: CNOT count for control structure in PREPARE-SELECT.

| Component | CNOT gates (sim.) | CNOT gates (hardware) |
|---|---|---|
| Routing circuit ($n_R=2$, $B_R=1$, $L=2$) | 3 | 3 |
| Single DRU expert ($n_D=4$, $B_E=8$, $L=2$) | 40 | $\sim$80 (controlled) |
| Coherent agg. ($n_R+n_D=6$, $L_{\text{agg}}=2$) | 12 | 12 |
| **Total (top-1, hardware)** | | **$\sim$95** |
| Error budget at $p=0.01$, 95 gates | | $\approx 62\%$ fidelity |
| Error budget at $p=0.001$, 95 gates | | $\approx 91\%$ fidelity |

At $p$=0.01, the hardware-equivalent circuit depth results in approximately 62% state fidelity under a conservative independent depolarising noise model, which is below the threshold for reliable inference. At $p$=0.001 - representative of leading superconducting devices (Bharti et al., 2022) - the fidelity rises to approximately 91%, making hardware deployment viable. The primary barrier to near-term hardware execution is therefore the two-qubit gate overhead of the controlled SELECT structure, not the coherent aggregation circuit itself. Reducing this overhead - for example through approximate circuit compilation (Shende et al., 2006) or native hardware-efficient ansatz designs - is a concrete direction for future work.

**Circuit depth and qubit count.** The routing circuit on $n_R$=2 qubits with $L$=2 variational layers and $B_R$=1 encoding block has a total depth (counting two-qubit gate layers) of approximately $2B_R + 2L = 6$ CNOT-ring layers, each of depth $\lceil n_R/2 \rceil = 1$ for the ring on 2 qubits. Each DRU expert on $n_D$=4 qubits with $B_E$=8 encoding blocks and $L$=2 variational layers per block has a depth of approximately $2(B_E + L) = 20$ CNOT-ring layers, each of depth $\lceil n_D/2 \rceil = 2$, totalling roughly 40 two-qubit gates. The aggregation circuit on $n_R + n_D = 6$ qubits with $L_{\text{agg}} = 2$ layers has approximately 12 two-qubit gates. All four experts are evaluated on separate qubit subregisters and can therefore run in parallel on hardware with sufficient qubit count.

**Native gate decomposition.** The Pauli rotation gates $R_X(\theta)$, $R_Y(\theta)$, $R_Z(\theta)$ are native or near-native on most superconducting and trapped-ion platforms (e.g., IBM Falcon/Eagle, IonQ Aria) (Krantz et al., 2019). The CNOT (CX) gate is the standard two-qubit entangling gate on IBM devices. The $R_Y$ gates in the aggregation circuit are single-qubit and introduce no additional two-qubit gate overhead.

**Parallelism and qubit layout.** With $K$=4 experts running in parallel, the total qubit requirement for a single forward pass is $K \cdot n_D + n_R = 4 \cdot 4 + 2 = 18$ qubits plus the aggregation register (shared with the routing-data joint state, 6 qubits). In the sparse top-$k$=1 setting, only one expert is evaluated per input, reducing the live qubit count to $n_D + n_R = 6$ data/routing qubits plus the aggregation register. This is within the qubit budget of current mid-scale NISQ devices (Bharti et al., 2022).

**Noise considerations.** Two-qubit gate error rates on current superconducting devices are in the range $10^{-3}$-$10^{-2}$ (Bharti et al., 2022), motivating our choice of $p$=0.01 for the simulation experiments. Our depolarising noise model (Section 5) is applied after each CNOT-ring layer during both training and evaluation, which is a standard and conservative simulation of gate-level noise (Nielsen & Chuang, 2010; Urbanek et al., 2021). The relatively shallow individual circuits (expert depth $\sim$40 two-qubit gates, routing depth $\sim$6) keep the cumulative noise below the threshold at which depolarising noise is known to induce exponential gradient concentration (Larocca et al., 2025) at $p$=0.01.

