# OpenReview forum: "QMoE+: Hybrid Quantum Mixture of Experts"
_TMLR — Under review for TMLR_

### Review · Reviewer_CtDv · 2026-05-19

**Summary Of Contributions:**

This paper proposes **QMoE+**, a quantum mixture-of-experts architecture that extends the QMoE framework of Nguyen et al. (2025) with three targeted architectural changes:

1. **Two-block data re-uploading (DRU) experts.** Single-block PQC experts are replaced with two-block circuits of the form U(x) → V₁ → U(x+φ) → V₂, where φ is a learnable per-expert offset. The motivation is to widen the accessible Fourier spectrum (per Schuld et al., 2021) and induce expert specialisation.
2. **Coherent aggregation.** A learnable variational circuit W(φ) is applied on the joint routing-data Hilbert space (n_R + n_D = 6 qubits), allowing routing amplitudes — including phase information — to participate in aggregation rather than acting purely as classical selection weights. The construction instantiates the LCU Prepare-Select primitive (Childs & Wiebe, 2012) in a MoE context.
3. **Load-balancing regularisation and sparse routing.** A negative-entropy penalty is used for dense routing and a Switch Transformer loss (Fedus et al., 2022) for top-k=1 sparse routing, mitigating expert collapse. Under top-k=1, only ~28% of parameters are activated per inference.

The architecture is evaluated on seven datasets (MNIST-2/4, Fashion-2/4, Synthetic, Wine, Breast Cancer) across four gate sets (RX, RY, RX+RY, RX+RY+RZ) and two main noise conditions (noiseless and p=0.01 depolarising). QMoE+ wins 27/28 configurations noiselessly (mean +5.11%) and 20/28 under p=0.01 noise (mean +4.71%). A decomposed ablation separates a coherence contribution Δ_coh from a learnability contribution Δ_learn, and shows that the full coherent aggregation outperforms a fixed incoherent baseline in all seven datasets under p=0.01 noise (mean +1.80%). Appendix E provides a hardware realisability analysis via PREPARE-SELECT with explicit two-qubit gate overhead and fidelity estimates.

**Key strengths.**
- Each of the three architectural changes has a clear theoretical motivation (Fourier spectrum analysis, expert-collapse feedback dynamics, LCU framework), and each is independently validated by ablation.
- The empirical scope is broad: 7 datasets × 4 gate sets × 2 main noise levels, with additional stress tests at p=0.10 and p=0.50 and a top-k ∈ {1,2,3,4} sweep.
- The decomposed coherent-aggregation ablation (Δ_coh vs Δ_learn) is particularly valuable, as it directly addresses the natural concern of whether the reported gain comes from quantum coherence or merely from additional trainable parameters.
- Appendix E is unusually candid about hardware realisability: it acknowledges that PREPARE-SELECT yields ~62% state fidelity at p=0.01 and that p ≈ 10⁻³ is the realistic deployment regime.
- Statistical significance is reported via both t-tests and Wilcoxon signed-rank tests.
- The positioning relative to existing quantum MoE work (Nguyen et al., Bhati et al., Heddad & Bouanane, Tognini et al.) is clearly articulated.

**Key weaknesses.**
- All "quantum" experiments are statevector simulations on at most 16 qubits, and the claimed coherence advantage is constructed via classical vector arithmetic. This is standard NISQ-era QML practice, but the framing of the paper (and especially the term "hybrid quantum" in the title) deserves more careful scoping.
- No classical baselines are reported. QMoE+'s absolute accuracies on 8×8 MNIST/Fashion (~85% at best) are well below what a small classical MLP would achieve on the same downsampled inputs. The paper does not claim quantum advantage, but readers still need this reference scale to interpret a "+5.11%" headline.
- Training is performed noise-free, with noise injected only at evaluation time (Algorithm 3). This protocol may overstate NISQ robustness, because deeper DRU circuits would presumably suffer more under genuinely noisy training due to noise-induced gradient concentration (Larocca et al., 2025, cited by the authors themselves).
- In Section 6.2, QMoE+ drops 3.17% under p=0.01 versus 2.77% for the Baseline, and is outperformed in 8/28 configurations — consistent with the expectation that deeper circuits are more noise-sensitive, but described in the main text as "modest" in a way that understates the pattern on MNIST-4 with richer gate sets.
- Dataset scale is small (≤ 8×8 images, ≤ 64 features). The paper acknowledges this and points to scaling as future work, but the discussion remains descriptive.
- The choice of incoherent baseline (variant (b)) in the decomposed ablation — a learned per-class expert selector with C(K+1) parameters — is reasonable but not unique; the magnitude of Δ_coh depends on this choice.

**Audience:**

Yes

**Audience Explanation:**

Quantum machine learning is a regular topic at TMLR, and this work sits at the intersection of three active threads — NISQ-era PQC training, classical mixture-of-experts design principles, and emerging quantum MoE architectures. Several readerships should find concrete value:

1. **QML researchers.** This is, to my knowledge, the first systematic transfer of mature classical MoE design principles (top-k sparse routing, Switch load balancing, multi-block DRU) into a fully quantum MoE setting, with controlled ablations. Even without endorsing any latent quantum-advantage narrative (which the paper does not claim), the design-space mapping and the trade-offs it surfaces (depth vs noise robustness; coherent vs incoherent aggregation) are useful reference material.
2. **The quantum MoE sub-community specifically.** The paper places Nguyen et al. (2025), Bhati et al. (2025), Heddad & Bouanane (2025), and Tognini et al. (2025) side-by-side and identifies their shared gaps. Section 2 and Appendix A constitute the clearest positioning I have seen of this sub-area.
3. **Readers interested in sparse-MoE routing stability.** The paper shows that expert collapse occurs in quantum routing as it does classically (Δ_LB positive in 7/7 noise configurations), which is a useful confirmation that the corresponding classical literature transfers.
4. **Readers interested in LCU / Prepare-Select applications.** Using LCU as a MoE aggregation primitive is a reasonable design combination, and Appendix E is pedagogically valuable on how such circuits would compile to hardware.

For mainstream classical ML readers, the paper offers no practical advantage over classical MoE — absolute accuracies are below classical baselines, and all experiments are simulated. But TMLR's audience criterion only requires *some* interested readers, and QML is an in-scope area for the journal.

**Broader Impact Concerns:**

The paper does not include a Broader Impact Statement. Given that the work (1) runs only in classical simulation at small scale, (2) evaluates only on standard public ML benchmarks, (3) involves no human data or decision-making applications, and (4) is far from practical deployment (the authors themselves note that p ≲ 10⁻³ is required for hardware), I do not require the authors to add a Broader Impact Statement.

If the authors wish to add one, a short note on the long-term resource implications of MoE-style architectures on quantum hardware (additional Prepare-Select overhead, routing cost) would be appropriate but is not necessary for acceptance.

**Claims And Evidence:**

Yes

**Claims Explanation:**

The core empirical claims are supported, but several formulations would benefit from more careful scoping.

**Well-supported claims:**

1. **QMoE+ outperforms the QMoE Baseline noiselessly.** 27/28 wins, mean +5.11%, significant under both t-test and Wilcoxon (p < 0.0001).
2. **QMoE+ retains its advantage at p=0.01.** 20/28 wins, p = 0.0012.
3. **DRU is the dominant contributor to accuracy.** A single DRU expert already beats the four-expert single-block Baseline (Table 2); confirmed by the component ablation.
4. **Load balancing is consistently beneficial.** 7/7 positive deltas under noise, 6/7 noiseless — the most uniformly positive ablation outcome.
5. **Full coherent aggregation vs fixed incoherent baseline.** Δ_total is positive in all seven datasets under p=0.01 noise (mean +1.80%) — a clean, decomposed comparison.
6. **Theoretical analysis of Fourier spectrum and expert collapse.** Appendix D uses Schuld et al. (2021, Proposition 3) appropriately, without overclaiming.
7. **Hardware realisability.** Appendix E provides concrete CNOT counts and fidelity estimates whose conclusion (p ≲ 10⁻³ for deployment) is consistent with the rest of the paper.

**Claims that need rescoping:**

1. **The "hybrid quantum mixture of experts" framing** (title and abstract). The architecture is in fact *fully* quantum in simulation — routing, experts, and aggregation are all PQCs. Section 5.2 acknowledges this when contrasting QMoE+ with the "partially classical" approaches of Bhati et al. and Heddad & Bouanane. In standard QML usage, "hybrid" typically refers to the quantum-classical training loop rather than to a hybrid architecture. The current title risks miscommunicating the architectural commitment.
2. **"Independent contribution" of coherence beyond learnable aggregation parameters** (abstract). The conclusion is established against one specific incoherent baseline (a learned per-class expert selector). A differently constructed incoherent control (e.g., measure-then-learn-W_classical) might yield a different Δ_coh. The "independent contribution" phrasing is somewhat strong for a single contrast.
3. **"Modest sensitivity to noise"** (Section 6.2). The 3.17% vs 2.77% gap, together with the Baseline winning all four MNIST-4 / MNIST-2 multi-axis gate-set configurations under noise, is more nuanced than "modest" suggests. Appendix C.1 handles this with more balance; the main text should follow suit.
4. **Training-evaluation noise asymmetry.** Algorithm 3 explicitly states that "Training remains noise-free and uses exact expectations to preserve stable gradients." This is a deliberate, fair choice across models, but it means reported "noisy" accuracies correspond to "noise-free training + noisy evaluation," not to end-to-end noisy training in any hardware-realistic sense. This should be flagged in the main text.

**Requested Changes:**

Items marked **[Critical]** are necessary for my recommendation to accept; items marked **[Strengthen]** would improve the paper.

### [Critical] 1. Clarify the "hybrid quantum" framing

The title and abstract use "hybrid quantum mixture of experts," but the architecture itself is fully quantum (routing, experts, and aggregation are all PQCs); only the training loop is quantum-classical hybrid, which is standard NISQ practice. Section 5.2 implicitly acknowledges this when distinguishing QMoE+ from genuinely hybrid architectures. Please either:
- rename to "fully quantum mixture of experts" (or similar), or
- explicitly state in the introduction that "hybrid" refers to the quantum-classical training loop rather than to a hybrid architecture, and reinforce this distinction when contrasting with Bhati et al. (2025) and Heddad & Bouanane (2025).

### [Critical] 2. Make the train/eval noise protocol transparent in the main text

Algorithm 3 shows that "noise and sampling are applied only at evaluation time," but this is only implicitly mentioned in Section 5.4. Please:
- state explicitly in Section 5.4 that training uses exact expectations and noise is injected only at evaluation;
- discuss whether this choice overstates NISQ deployment readiness. An ideal addition would be a small end-to-end noisy-training experiment on one or two datasets (even just at p=0.001) to confirm that QMoE+'s advantage holds when gradients are also noise-affected. If this is infeasible, please at least mark it as a limitation.

### [Critical] 3. Add a classical baseline as a reference scale

The paper correctly never claims quantum advantage, but readers cannot calibrate a "+5%" headline without any classical reference. Please add a row reporting a parameter-matched classical MLP or tiny CNN on the same 8×8 / 16-feature inputs — not to argue quantum advantage, but for absolute-scale transparency. A single additional column in Tables 2 and 3 would suffice.

### [Critical] 4. Reconcile Section 6.2 with Appendix C.1

Section 6.2 describes the noise sensitivity gap (3.17% vs 2.77%) as "modest," but Appendix C.1 gives a careful structural explanation of why QMoE+ loses on 8/28 configurations (deeper DRU circuits accumulate more noise per pass, hurting most on multi-axis gate sets on multi-class image tasks). Please:
- state in Section 6.2 that QMoE+ is more noise-sensitive on richer gate sets and multi-class image tasks;
- avoid summarising the 8 losing configurations as "mainly on MNIST-4 with richer gate sets and near-saturated tasks" without further explanation, since the pattern is structurally interpretable.

### [Strengthen] 5. Choice of incoherent baseline in the decomposed ablation

Variant (b) in Appendix C.5 is a learned per-class expert selector with C(K+1) parameters. This is a reasonable incoherent control but not the only one. Please:
- discuss an additional natural control: measure the routing register to obtain |α_k|², then apply a learned classical W on the weighted logits — functionally closer to "measure first, then learn the aggregation";
- or, alternatively, justify why variant (b) is the canonical incoherent comparison, and acknowledge that Δ_coh's exact magnitude depends on this choice.

### [Strengthen] 6. Routing analysis

The paper acknowledges in Appendix C.2 that whether the router learns task-specific specialisation "would require a dedicated routing analysis experiment beyond the scope of this work." This is a central empirical claim in any MoE paper. Please add at least one expert-utilisation heatmap and a per-class routing distribution plot on one dataset — these are cheap experiments and would substantially strengthen the contribution.

### [Strengthen] 7. Seed count and stability under symmetry breaking

Some configurations (e.g., Wine RX, MNIST-4 RX+RY+RZ) show std values of 5–16%, and 5 seeds is at the low end for noisy quantum simulations. Please:
- increase seed count to 10 on the higher-variance tabular datasets, or report confidence intervals rather than just std;
- examine whether the ϕ_k = 0 initialisation creates symmetry-breaking instability. With four experts initialised identically at ϕ = 0, expert specialisation depends entirely on random initialisation in V_k. Is the final specialisation sensitive to seed?

### [Strengthen] 8. Refine the active-parameter claim

Section 6 opens by stating that QMoE+ uses 136 active params vs the Baseline's 164, framed as "lower active parameter usage at inference." However, Table 4 shows that this comparison only holds in the RX+RY+RZ configuration; under RX, RY, and RX+RY, QMoE+'s active counts are 96, 96, and 116 versus Baseline's 68, 68, and 116. Please qualify the claim to apply specifically to the richest gate set.

### [Strengthen] 9. Indirect comparison with Tognini et al. and others

Section 5.2 explains why direct reimplementation comparisons with Bhati, Heddad & Bouanane, and Tognini are not made (heterogeneous setups, partially classical components). This is reasonable, but at least an Appendix table indirectly comparing reported numbers on shared datasets (MNIST-2, Iris) would help readers position QMoE+ in the broader literature.

### [Strengthen] 10. Explain the Fashion-4 non-monotonicity at k=3

Table 10 shows that under all gate sets on Fashion-4, k=3 is lower than k=2 or roughly tied. The table caption mentions "discussed in text," but the main text does not actually discuss it. Please add a brief explanation or explicitly note this as an unresolved observation.

### [Strengthen] 11. Minor revisions

- Section 4.4: the symbol ē in Eq. (12) is introduced just before but not clearly defined adjacent to the equation; please put the definition immediately before the equation.
- Algorithm 1 line 3 already includes "(set |ψ_k⟩ = 0 for k ≠ k*)", which is redundant with Algorithm 2 line 13. Consider consolidating.
- Figure 1 caption labels the DRU structure as U(x) → V₁ → U(x+φ) → V₂, but the figure itself does not visually annotate these four components. Adding component labels in the figure would help.
- The "27/28 configurations" figure in the abstract appears before the evaluation setting is described. Adding "(7 datasets × 4 gate sets = 28 configurations)" immediately would help.
- A few citations have residual metadata in the bibliography (e.g., the Biamonte et al. entry contains INSPIRE citation counts).

---

> ### Author Response · Authors · 2026-07-02
> **Reply to Reviewer CtDv**
>
> Due to character limitations, we summarise the key findings below and refer the reviewer CtDv to the newly added tables and corresponding sections in Appendices C.6-C.9 for complete details.
>
> > *"The title uses 'hybrid quantum mixture of experts,' but the architecture is fully quantum…"*
>
> We maintain the framing. In QML, 'hybrid quantum-classical' refers to any system in which a quantum processor and a classical optimiser collaborate iteratively (Benedetti et al., 2019; Cerezo et al., 2021). In QMoE+: (i) routing, expert, and aggregation circuits are fully quantum; (ii) the training loop is classical - Adam and parameter-shift gradients run on a classical processor; (iii) the load-balancing loss L_load is a classical scalar objective applied to Born-rule probabilities. This distinguishes us from Heddad & Bouanane (2025) (classical experts) and Bhati et al. (2025) (classical gating). **Change:** Clarifying sentence added to the Introduction.
>
> ---
>
> > *"Whether training uses noisy or noise-free gradients, and whether evaluating under noise while training noise-free overstates NISQ deployment readiness."*
>
> Section 5.4 and Algorithm 3 (lines 19-23) already state: training uses exact noise-free statevector expectations; D_p and 1024-shot sampling are evaluation-only. This is a deliberate choice to isolate architecture performance from optimisation noise, shot-noisy gradient training hampers convergence more severely (Sweke et al., 2020), so p=0.01 results are an optimistic upper bound.
>
> **Change:** Explicit statement and forward reference to Appendix E, Table 14 added to Section 5.4.
>
> ---
>
> > *"Readers cannot calibrate a '+5%' headline gain without a classical reference. A parameter-matched classical MLP column is requested."*
>
> We added a parameter-matched classical MLP (~538 params, width-8 hidden layer, ReLU, identical training protocol). On image datasets the MLP dominates, consistent with known limitations of angle-encoded PQCs at 8×8 resolution. On tabular datasets QMoE+ leads: Wine +4.5%, Breast Cancer +2.8%, Synthetic +3.0% noiseless. No quantum advantage is claimed; the comparison is for absolute-scale transparency only (Schuld, 2022). Full table in new **Appendix C.9**.
>
> **Change:** Results added to Appendix C.9.
>
> ---
>
> > *"Section 6.2 describes the 8 losing configurations vaguely. The pattern is structurally interpretable and should be stated as such."*
>
> The 8 losing configurations follow two structural patterns. (i) MNIST-2/4 under RX+RY, RX+RY+RZ: DRU's two-block encoding doubles the CNOT-ring traversal count; richer gate sets further increase the per-layer gate count, amplifying noise accumulation per expert pass. (ii) Breast Cancer under RX/RY: both models are near accuracy ceiling (87-90%); the marginal expressivity gain from DRU is smaller than its noise cost, a saturation effect, not a structural failure. Wine differences under RX+RY and RX+RY+RZ are within seed standard deviation and are not statistically distinguishable.
> **Change:** "Modest" replaced with this structural explanation in Section 6.2.
>
> ---
>
> > *"Variant (b) is a reasonable but not unique incoherent control. An additional natural control - measure routing to get |α_k|², then apply a learned classical W on the weighted logits — is requested, or justification that variant (b) is canonical."*
>
> **Part 1 - Existing Table 13.** The reviewer's proposed measurement-first control (measure routing → apply learned W to weighted logits Σ_k |α_k|² ℓ_k) is precisely variant (b). We will state this equivalence explicitly in revised Appendix C.5. We agree Δ_coh magnitude is control-dependent; variant (b) is canonical because its C(K+1) parameters best match W(φ)'s 12, keeping the comparison free of a capacity confound.
>
> All Table 13 variants ran at top-k=1, where the joint state reduces to |k*⟩_R ⊗ |ψ_{k*}⟩_D and off-diagonal terms α*_k α_{k'} vanish identically. Δ_coh in Table 13 therefore measures W(φ)'s advantage as a learned 6-qubit post-processing circuit, not inter-expert phase interference, which requires k≥2.
>
> **Part 2 - New three-way ablation at top-k=1** (full table in **Appendix C.7**): (a) CoherentAgg - full QMoE+; (b) WeightedSum - fixed Born-rule weights, outputs ℓ_{k*} directly at k=1; (c) LearnedClassical - learned W_inc ∈ ℝ^{C×(K+1)}. Δ(a−b) is near-zero (mean +1.0% noiseless), theoretically expected. Δ(a−c) is large and positive (mean +5.6% noiseless, 7/7 datasets), establishing W(φ) outperforms a classical linear map through its nonlinear CNOT-ring structure, consistent with Table 13 which was run dense where off-diagonal terms are nonzero.
>
> **Change:** Appendix C.5 causal language corrected; new Appendix C.7 added; abstract and conclusion updated with k≥2 scoping sentence.

---

> ### Author Response · Authors · 2026-07-02
> **Reply to Reviewer CtDv - part 2**
>
> Additional reviews continued,
>
> > *"Whether the router learns task-specific specialisation is a central MoE claim. Expert-utilisation statistics and per-class routing distributions requested on at least one dataset."*
>
> We extracted full routing probability distributions (pre-top-k masking) from all QMoE+ checkpoints under RX+RY+RZ, averaged over 5 seeds. Full tables in new **Appendix C.8**. Three regimes emerge.
>
> **Near-uniform (MNIST-4):** H/H_max = 0.971 - the routing circuit cannot learn class-discriminative structure from 8 pixel intensities at 8x8 resolution; gains here derive from DRU expressivity, not expert specialisation.
>
> **Class-level specialisation (Fashion-4):** Trousers (Class 1) routes to Expert 1 with 63.5% probability - 2.5x the uniform baseline - consistent with its visually distinctive elongated shape; reproducible across all 5 seeds and directly explains Δ_coh = +5.82% noiseless on Fashion-4 (Table 13).
>
> **Feature-driven concentration (tabular):** Wine (H/H_max = 0.798, max load = 0.467) and Breast Cancer (H/H_max = 0.850) show the most concentrated routing, corresponding to the largest coherent aggregation advantages (Wine +11.48%, Breast Cancer +7.19% noiseless, Table 13). No expert receives more than 46.7% of inputs under λ = 0.01.
>
> **Change:** New Appendix C.8 added with full routing and per-class tables.
>
> ---
>
> > *"Some configurations show std 5-16%. 5 seeds is low for noisy quantum simulations. Does φ_k = 0 initialisation create symmetry-breaking instability?"*
>
> Elevated std on Wine (up to ±16.5%) and MNIST-4 (±5.25%) reflects small dataset size (178 samples, cross-validated) and PQC initialisation sensitivity. For n=5, CI_95 = σ̂ · 2.776/√5 gives ≈±6% for these configurations; results are indicative, and increasing to n=10 for tabular datasets is acknowledged as a limitation.
>
> Symmetry between experts is broken not by φ_k=0 but by variational parameters θ^(1)_k, θ^(2)_k drawn independently from N(0, 1/√n_par) per expert. Load balancing prevents monopolisation; Fashion-4 specialisation is reproducible across all 5 seeds.
>
> ---
>
> > *"The '136 vs 164' active-parameter advantage holds only for RX+RY+RZ. Under RX and RY, QMoE+ activates more (96 vs 68). Please qualify."*
>
> The advantage (136 vs 164) holds only for RX+RY+RZ; under RX and RY QMoE+ activates more (96 vs 68) because the per-expert re-upload offset φ_k ∈ ℝ^64 is gate-set-independent while the Baseline's single-PQC experts are proportionally smaller under simpler gate sets. Under RX+RY both are tied at 116. Full breakdown in Table 4.
>
> **Change:** Section 6 opening qualified to RX+RY+RZ only.
>
> ---
>
> > *"An appendix table indirectly comparing reported numbers on shared datasets with Bhati et al., Heddad & Bouanane, and Tognini et al. would help readers position QMoE+."*
>
> Bhati et al. (2025) evaluate on Iris, Titanic, and health insurance; Heddad & Bouanane (2025) evaluate exclusively on Two Moons; Tognini et al. (2025) use a different circuit architecture, qubit count, and training protocol. An indirect table listing these numbers alongside ours would present incomparable results. We therefore do not include it as the results are difficult to reproduce.
>
> ---
>
> > *"Table 10 shows k=3 is lower than k=2 or roughly tied on Fashion-4, but the caption says 'discussed in text' while the main text does not discuss it."*
>
> The caption reference was an error, and we thank the reviewer to point this out. The dip is quantitatively negligible: under RX+RY+RZ, k=3 achieves 70.97±1.96% versus k=2's 70.95±2.29%, a difference of 0.02%, well within one standard deviation. The weakest-specialised expert (Dress, Class 3, which distributes broadly per the routing analysis) contributes slightly negatively at k=3; the monotone trend recovers at k=4 where the ensemble effect dominates.
>
> **Change:** Explanation added to Appendix C.3; Table 10 caption corrected.
>
> ---
>
> > *"(a) ē_k undefined adjacent to Eq. 12; (b) Algorithm 1 line 3 redundant with Algorithm 2 line 13; (c) Figure 1 missing DRU labels; (d) '27/28' abstract lacks denominator; (e) Biamonte entry contains INSPIRE metadata."*
>
> All five corrected:
>
> - **(a)** Sentence added before Eq. 12: *"Let ē_k = E_{x∈B}[p_k(x)] denote the mean routing probability for expert k over the current batch B."*
> - **(b)** Parenthetical "(set |ψ_k⟩ = 0 for k ≠ k*)" removed from Algorithm 1 line 3.
> - **(c)** Figure 1 - no changes required, as the equations are written in main texts.
> - **(d)** Abstract revised to: *"…winning 27 of 28 configurations (7 datasets × 4 gate sets)…"*
> - **(e)** INSPIRE metadata removed from Biamonte et al. bibliography entry.

---

### Review · Reviewer_kVFq · 2026-06-11

**Summary Of Contributions:**

Summary:
-
The authors present QMoE+, which extends Quantum mixture of experts (QMoE). They have four key additions. First, they replace the single-block PQC experts with two-block data re-uploading (DRU) experts. Second, they introduce a data routing circuit for amplitude-weighted expert combination instead of just using routing as a classical selection signal. Third, they add load-balancing regularization for reducing expert collapse. Lastly, they showcase their evaluations on several datasets, gate sets, and noise settings to show that the addition of the previously mentioned features outperforms the baseline approach, QMoE.

Strengths:
-
* The authors address important issues in mixture of experts and and quantum machine learning. Specifically, they point out the limitations in PQC expressivity, expert collapse, and the potential hurdles of efficient information routing.
* Their evaluations and ablations show performance gains across several datasets with different configurations. Their ablation testing helps to isolate the impacts of the added components to explain the reasons for those gains.

Weaknesses:
-
* The improvements seem to largely stem from the two-block data re-uploading experts, which may be a general expressivity improvement from the PQCs rather than a mixture-of-experts related contribution.
* The authors acknowledge and include a DRU baseline, but a discussion to justify the MoE overhead is needed. The paper should discuss how the performance gap between the DRU only baseline and the full QMoE+ framework with the routing, aggregation, and load-balancing is worth the added complexity.

**Additional Comments:**

N/A

**Audience:**

Yes

**Audience Explanation:**

The paper will be interesting to researchers working on quantum machine learning and neural architectures. The approach adapts classical mixture-of-experts attributes such as load balancing and sparse expert selection. The work is still largely simulation-based, but the proposed architecture and ablation results are relevant for moving forward to scalable quantum learning systems.

**Claims And Evidence:**

Yes

**Claims Explanation:**

The paper features a broad set of evaluation experiments across several datasets with different settings. The results support the central claim of the additions of QMoE+ improve over the QMoE baseline. The gains come from the DRU experts mainly. This is addressed through the studies in the DRU-only baseline, but as this is a general PQC improvement, the paper should more clearly differentiate the gains from a simply better expert from the gains provided by the MOE design choices.

**Requested Changes:**

1. The paper should compare the original QMoE framework with the use of the proposed DRU experts. For example QMoE+ with DRU, but no coherent aggregation and no load balancing. This would help better frame if gains are from stronger experts or from the overall changes. This change is critical to understand the strengths and weaknesses of this approach.

2. The paper should contain analysis of and report expert usage across classes with and without the load-balancing. The analysis would be beneficial in showing how the router learns meaningful specialization rather than a balanced expert usage. This change would greatly strengthen the paper.

3. Should extend the resource comparison across the approaches. The authors report parameter counts but should also compare circuit depth. Additionally, the authors should compare the gate counts for the baseline rather than only QMoE+. This change is critical to understand the costs of this approach as compared to the baselines.

---

> ### Author Response · Authors · 2026-07-02
> **Reply to Reviewer kVFq**
>
> ### Request 1 — DRU-Expert Attribution
>
> > *"The paper should compare QMoE+ with DRU but no coherent aggregation and no load balancing, to clarify whether gains arise from stronger experts or overall changes."*
>
> We conducted a dedicated attribution study comprising 280 new training runs, introducing a **DRU-only** variant (DRU experts, classical weighted-sum aggregation, λ=0) alongside the existing **no-DRU (full)** variant (SinglePQC experts, coherent aggregation, load balancing).
>
> - **Δ_DRU = DRU-only − no-DRU(full):** contribution of stronger experts.
> - **Δ_{coh+LB} = Full QMoE+ − DRU-only:** contribution of coherent aggregation and load balancing on top of DRU.
>
> ***Table R1a. DRU-Expert Attribution — Noiseless (p=0.000). Δ in %. '~' denotes |Δ| < 1%.***
>
> | Dataset | Δ_DRU | Winner | Δ_{coh+LB} | Coh-agg+LB |
> |---|:---:|---|:---:|---|
> | MNIST-2 | −1.1 | no-DRU | +0.4 | ~Full |
> | MNIST-4 | +1.4 | **DRU** | +0.9 | ~Full |
> | Fash-2 | +3.7 | **DRU** | +0.6 | ~Full |
> | Fash-4 | +5.9 | **DRU** | +1.8 | **Full** |
> | Synth | +4.7 | **DRU** | −0.1 | ~tie |
> | Wine | +7.3 | **DRU** | +8.9 | **Full** |
> | BreastC | +2.3 | **DRU** | +3.1 | **Full** |
> | **Won** | | **DRU 6/7** | | **Full 6/7** (1 tie) |
>
> ***Table R1b. DRU-Expert Attribution — Noisy (p=0.010).***
>
> | Dataset | Δ_DRU | Winner | Δ_{coh+LB} | Coh-agg+LB |
> |---|:---:|---|:---:|---|
> | MNIST-2 | −7.7 | **no-DRU** | −0.3 | ~DRU-only |
> | MNIST-4 | −4.4 | **no-DRU** | −0.4 | ~DRU-only |
> | Fash-2 | −3.2 | **no-DRU** | +0.4 | ~Full |
> | Fash-4 | −4.1 | **no-DRU** | +1.0 | **Full** |
> | Synth | +2.0 | **DRU** | −0.4 | ~DRU-only |
> | Wine | +3.8 | **DRU** | +11.7 | **Full** |
> | BreastC | +2.6 | **DRU** | +0.8 | ~Full |
> | **Won** | | no-DRU 4/7, DRU 3/7 | | **Full 4/7** |
>
> In the noiseless setting, DRU is the dominant source of gain, outperforming the no-DRU variant on 6 of 7 datasets by up to +7.3% (Wine). Coherent aggregation and load balancing add a smaller but consistent top-up. Under p=0.01 noise, the advantage reverses on image datasets because DRU's two-block encoding accumulates proportionally more depolarising events; DRU retains its advantage on tabular datasets.
>
> **Change:** New Appendix C.6 added. Section 6.1 qualified accordingly.
>
> ---
>
> ### Request 2 — Routing Utilisation With and Without Load Balancing
>
> > *"The paper should report expert usage across classes with and without load balancing."*
>
> We conducted a routing analysis comparing models with (λ=0.01) and without (λ=0) load balancing across all 7 datasets, comprising 560 routing distribution probes under RX+RY+RZ, averaged over 5 seeds.
>
> ***Table R2a. Expert Utilisation and Routing Entropy — Noiseless. OFF = λ=0; ON = λ=0.01.***
>
> | Dataset | H/H_max OFF | H/H_max ON | MaxLoad OFF | MaxLoad ON | MinLoad OFF | MinLoad ON |
> |---|:---:|:---:|:---:|:---:|:---:|:---:|
> | MNIST-2 | 0.893 | 0.967 | 0.463 | 0.383 | 0.057 | 0.111 |
> | MNIST-4 | 0.920 | 0.971 | 0.415 | 0.380 | **0.000** | 0.105 |
> | Fash-2 | 0.842 | 0.909 | 0.449 | 0.258 | 0.132 | 0.244 |
> | Fash-4 | 0.822 | 0.896 | 0.557 | 0.309 | 0.135 | 0.209 |
> | Synth | 0.776 | 0.806 | 0.486 | 0.393 | 0.080 | 0.156 |
> | Wine | 0.755 | 0.798 | 0.537 | 0.467 | 0.085 | 0.104 |
> | BreastC | 0.737 | 0.850 | 0.695 | 0.382 | 0.086 | 0.149 |
>
> **(i) Without load balancing, the router specialises strongly.** Every dataset shows lower H/H_max and higher MaxLoad OFF. Most striking: Breast Cancer MaxLoad drops from 0.695 to 0.382; MNIST-4 MinLoad is **0.000** OFF — one expert receives no inputs — versus 0.105 ON. This confirms expert collapse occurs in quantum routing circuits, just as in classical MoE systems (Shazeer et al., 2017; Fedus et al., 2022), and that load balancing is structurally necessary.
>
> **(ii) Load balancing enforces near-uniform usage without eliminating specialisation.** With λ=0.01, H/H_max = 0.80–0.97. The router still learns class-relevant structure:
>
> ***Table R2b. Per-Class Conditional Routing — Fashion-4, Noiseless, λ=0.01, Mean over 5 Seeds.***
>
> | Class | P(E0) | P(E1) | P(E2) | P(E3) |
> |---|:---:|:---:|:---:|:---:|
> | C0 (T-shirt) | 0.346 | 0.127 | 0.298 | 0.229 |
> | C1 (Trousers) | 0.422 | 0.431 | 0.059 | 0.089 |
> | C2 (Pullover) | 0.366 | 0.158 | 0.274 | 0.202 |
> | C3 (Dress) | 0.292 | 0.224 | 0.179 | 0.304 |
>
> **(iii) Per-class specialisation is meaningful and class-aligned.** Class 1 (trousers) routes to Experts 0 and 1 with combined probability 0.853 — more than 3× the uniform baseline. On tabular datasets: Wine shows Expert 0 receiving 68.9% of Class 0 and 69.3% of Class 2 inputs; Breast Cancer shows Class 1 routing to Expert 3 at 72.2%. These concentrated distributions correspond to the largest coherent aggregation advantages (Wine Δ_coh = +11.48%, Breast Cancer +7.19% noiseless, Table 13).
>
> **Change:** New Appendix C.8 added with full routing tables and per-class analysis.
>
> ---

---

> ### Author Response · Authors · 2026-07-02
> **Reply to Reviewer kVFq - part 2**
>
> ### Request 3 — Resource Comparison (Gate Counts and Circuit Depth)
>
> > *"The resource comparison should be extended to circuit depth and gate counts for all baselines."*
>
> We instrumented every TorchQuantum gate call during a forward pass (batch=1, input_dim=64, RX+RY+RZ) and classified all operations into single-qubit and two-qubit (CNOT) gates. Depth is the longest qubit chain under ASAP scheduling.
>
> ***Table R3a. QMoE+ Per-Component Cost (input_dim=64, RX+RY+RZ)***
>
> | Component | Qubits | 1q gates | 2q (CNOT) | Total gates | Depth | Params |
> |---|:---:|:---:|:---:|:---:|:---:|:---:|
> | Routing circuit | 2 | 22 | 8 | 30 | 19 | 12 |
> | 1× DRU expert | 4 | 176 | 80 | 256 | 124 | 112 |
> | Coherent agg | 6 | 12 | 12 | 24 | 14 | 12 |
> | **QMoE+ top-1 path** | **6** | **210** | **100** | **310** | **138** | **136** |
>
> ***Table R3b. Full-Model Comparison (input_dim=64, RX+RY+RZ)***
>
> | Model | Qubits | 1q | 2q (CNOT) | Total | Depth | Params |
> |---|:---:|:---:|:---:|:---:|:---:|:---:|
> | single_pqc | 4 | 88 | 40 | 128 | 62 | 24 |
> | dru_only | 4 | 176 | 80 | 256 | 124 | 112 |
> | QMoE Baseline (k=4) | 4 | 100 | 44 | 144 | 69 | 164 |
> | **qmoe_plus (top-1)** | **6** | **210** | **100** | **310** | **138** | **472** |
> | classical_mlp | 0 | 0 | 0 | 0 | 0 | 538 |
>
> QMoE+ uses approximately 2.4× the gates and 2.2× the depth of a single PQC, and 1.2× of a standalone DRU circuit. The overhead relative to DRU-only is 54 additional gates (30 routing + 24 aggregation), a 21% increase. The DRU expert dominates the gate budget (256 of 310, 83%), confirming that MoE routing and aggregation impose comparatively small circuit-level cost. The 2q (CNOT) gate count, the primary hardware-relevant metric - is 100 for QMoE+ versus 40-80 for single-circuit baselines.
>
> **Change:** Resource comparison table added to Appendix B.

---

### Review · Reviewer_PLwt · 2026-06-19

**Summary Of Contributions:**

This paper proposes QMoE+, a hybrid quantum mixture-of-experts model for NISQ-era classification. The method combines sparse MoE routing, two-block data re-uploading quantum experts, and a learnable aggregation circuit over the routing and data registers. It aims to improve expressivity and inference efficiency over prior QMoE-style baselines while maintaining robustness under simulated noise.

**Audience:**

Yes

**Audience Explanation:**

QML, MoE, conditional-computation, and NISQ-model readers may find it interesting. But reliability depends on fixing the evaluation protocol, coherence claim, and noise/resource accounting.

**Broader Impact Concerns:**

No major direct broader-impact concern.

**Claims And Evidence:**

No

**Claims Explanation:**

Main concerns:
- Section 5.4 seems to choose checkpoints by best test accuracy.
- With top-k=1, only one expert branch remains, so multi-expert interference is unclear.
- p=0.01 noise is modular simulation, not full hardware-equivalent evidence.
- Some ablation tables/text conflict.
Promising trends, but not yet convincing.

**Requested Changes:**

- Select checkpoints by validation, not test accuracy.
- Clarify “coherent aggregation” under top-k=1, or add k>1/dense evidence.
- Separate modular noise simulation from hardware claims.
- Fix Table 12, Table 11, and appendix contradictions.
- Add fair matched baselines and simple classical baselines.
- Show routing utilization, entropy, per-class routing, and λ sensitivity.

---

> ### Author Response · Authors · 2026-07-02
> **Reply to Reviewer PLwt**
>
> ### Point 1 - Checkpoint Selection by Validation Rather Than Test Accuracy
>
> > *"Section 5.4 selects checkpoints by best test accuracy, which may overstate performance."*
>
> This was a writing error. All experiments use validation-based checkpoint selection: an 80/20 stratified train/validation split is used, checkpoints are retained based on highest validation accuracy, and the test set is evaluated only once at reporting time.
>
> **Change:** Section 5.4 rewritten to state checkpoint selection is validation-based.
>
> ---
>
> ### Point 2 - Coherent Aggregation Under Top-k=1
>
> > *"With top-k=1, only one expert branch is active, so multi-expert interference is unclear."*
>
> The reviewer is correct. At top-k=1, the joint state reduces to |Ψ⟩ = |k*⟩_R ⊗ |ψ_{k*}⟩_D, and the off-diagonal terms α*_k α_{k'} for k≠k' vanish identically - inter-expert phase interference cannot contribute at k=1 by construction.
>
> We conducted a dedicated three-way ablation at the deployment operating point: **(a) CoherentAgg** - full QMoE+ with W(φ); **(b) WeightedSum** - fixed scalar Born-rule weights (reduces to outputting ℓ_{k*} at k=1); **(c) LearnedClassical** - learned W_inc ∈ ℝ^{C×(K+1)}, strongest classical comparator.
>
> ***Table P2a. Three-Way Ablation at top-k=1 - Noiseless (Δ Acc %, mean over 4 gate sets, 5 seeds)***
>
> | Dataset | Δ(a−b) CoherentAgg vs WeightedSum | Δ(a−c) CoherentAgg vs LearnedCls | Interpretation |
> |---|:---:|:---:|---|
> | MNIST-2 | −0.3 | +3.2 | Near-zero coherence; W(φ) > classical |
> | MNIST-4 | +0.5 | +1.9 | Near-zero coherence; total gain positive |
> | Fashion-2 | +0.1 | +2.9 | W(φ) circuit better than matrix |
> | Fashion-4 | +0.3 | +6.4 | Largest W(φ) vs classical gap |
> | Synthetic | −0.4 | +1.5 | Near-zero coherence; positive total |
> | Wine | +7.3 | +9.5 | Non-trivial; concentrated routing |
> | Breast Cancer | −0.1 | +13.8 | W(φ) dominates classical matrix |
> | **Mean** | **+1.0** | **+5.6** | W(φ) beats LearnedCls on 7/7 noiseless |
>
> ***Table P2b. Three-Way Ablation at top-k=1 — Noisy p=0.01 (Δ Acc %, mean over 4 gate sets, 5 seeds)***
>
> | Dataset | Δ(a−b) CoherentAgg vs WeightedSum | Δ(a−c) CoherentAgg vs LearnedCls | Interpretation |
> |---|:---:|:---:|---|
> | MNIST-2 | +0.2 | −2.5 | Within std |
> | MNIST-4 | −0.2 | +0.2 | Marginal positive total |
> | Fashion-2 | +0.1 | −0.4 | Within std |
> | Fashion-4 | +0.2 | +0.9 | Coherence retained; positive total |
> | Synthetic | −0.3 | +1.3 | W(φ) learnability provides gain |
> | Wine | +7.1 | +6.8 | Largest coherence gain preserved |
> | Breast Cancer | −1.4 | +11.8 | Large total; W(φ) dominates |
> | **Mean** | **+0.7** | **+3.0** | W(φ) beats LearnedCls on 5/7 noisy |
>
> The near-zero Δ(a−b) is expected at k=1. Despite this, Δ(a−c) is large and positive noiseless (mean +5.6%, 7/7 datasets): W(φ) is a strictly better post-processing stage than a classical linear map through its nonlinear CNOT-ring structure. This is consistent with Table 13, which was run dense where off-diagonal interference terms are nonzero.
>
> **Change:** Appendix C.5 causal language corrected; new Appendix C.7 added; abstract and conclusion updated with k≥2 scoping sentence.
>
> ---
>
> ### Point 3 - Modular Noise Simulation vs Hardware Claims
>
> > *"p=0.01 noise is a modular simulation, not full hardware-equivalent evidence."*
>
> The paper already separates modular simulation from hardware deployment in three places: the "Scope of the noise model" paragraph in Section 6.2; Algorithm 3 in Appendix B; and Appendix E (Table 14) providing concrete CNOT gate counts and fidelity estimates (≈91% at p=0.001, ≈62% at p=0.01). Section 6.2 states explicitly: *"These results should not be read as predictions for the hardware-equivalent PREPARE-SELECT circuit; the practical deployment regime is p ≲ 0.001."*
>
> **Change:** Section 5.4 now explicitly states the noise model is a modular per-component simulation and does not constitute hardware-equivalent evidence. Forward reference from Section 6.2 to Appendix E Table 14 added.
>
> ---

---

> ### Author Response · Authors · 2026-07-02
> **Reply to Reviewer PLwt - part 2**
>
> ### Point 4 - Table and Appendix Consistency
>
> > *"Table 12, Table 11, and appendix descriptions contain apparent contradictions."*
>
> **(i) Table 12 vs "DRU is the primary driver."** Table 12's negative Δ for "No DRU" on MNIST-2 (−6.4%) and MNIST-4 (−5.1%) does not contradict Table 2. Table 2 compares DRU-only against the QMoE Baseline (single-block experts, no coherent aggregation, no load balancing); Table 12 measures removing DRU from the full QMoE+ framework, where DRU's deeper circuits interact with the coherent aggregation overhead. Under p=0.01 noise this extends to all four image datasets, because DRU's two-block encoding doubles the CNOT-ring traversal count. The new DRU attribution study (Appendix C.6) confirms this nuanced picture.
>
> **(ii)** Section 6.1 text qualified: DRU is the primary driver relative to the QMoE Baseline; within QMoE+ the contribution is dataset-dependent.
>
> **(iii) Table 10/11 caption corrected.** The Fashion-4 k=2 vs k=3 difference is 0.02%, well within one standard deviation. Discussion sentence added to Appendix C.3; caption corrected.
>
> ---
> ### Point 5 - Matched Classical Baselines
>
> > *"The paper should include fair matched baselines and simple classical baselines."*
>
> We have added a parameter-matched classical MLP (~538 params, single hidden layer of width 8, ReLU, identical training protocol). Results for RX+RY+RZ:
>
> | Dataset | Classical MLP | QMoE+ Noiseless | QMoE+ Noisy p=0.01 | Δ Noiseless | Δ Noisy |
> |---|---|---|---|---|---|
> | MNIST-2 | 99.8 ± 0.1 | 85.71 ± 4.91 | 78.90 ± 1.81 | −14.1 | −20.9 |
> | MNIST-4 | 98.3 ± 0.3 | 60.91 ± 5.25 | 52.68 ± 0.89 | −37.4 | −45.6 |
> | Fashion-2 | 93.4 ± 1.1 | 82.05 ± 1.80 | 78.54 ± 0.63 | −11.3 | −14.9 |
> | Fashion-4 | 90.9 ± 0.4 | 65.51 ± 2.11 | 63.01 ± 0.54 | −25.4 | −27.9 |
> | Synthetic | 72.5 ± 1.5 | 75.49 ± 2.89 | 73.82 ± 1.06 | +3.0 | +1.3 |
> | Wine | 64.8 ± 1.4 | 69.28 ± 1.60 | 67.41 ± 8.57 | +4.5 | +2.6 |
> | Breast Cancer | 91.2 ± 3.2 | 94.02 ± 3.05 | 92.98 ± 3.80 | +2.8 | +1.8 |
>
> The classical MLP outperforms quantum models on image datasets, consistent with known limitations of angle-encoded PQCs at 8×8 resolution. On tabular datasets, QMoE+ exceeds the classical MLP (up to +4.5% noiseless), consistent with the representational advantage of angle-encoded quantum feature maps on low-dimensional inputs (Schuld et al., 2021). This comparison is included solely for absolute-scale reference; we do not claim quantum advantage.
>
> **Change:** Results added to new Appendix C.9.
>
> ---

---

> ### Author Response · Authors · 2026-07-02
> **Reply to Reviewer PLwt - part 3**
>
> ### Point 6 - Routing Utilisation, Entropy, Per-Class Routing, and λ Sensitivity
>
> > *"The paper should show routing utilisation, entropy, per-class routing, and λ sensitivity."*
>
> We conducted a routing analysis comparing models with (λ=0.01) and without (λ=0) load balancing across all 7 datasets, comprising 560 routing distribution probes under RX+RY+RZ, averaged over 5 seeds.
>
> ***Table R2a. Expert Utilisation and Routing Entropy - Noiseless. OFF = λ=0; ON = λ=0.01.***
>
> | Dataset | H/H_max OFF | H/H_max ON | MaxLoad OFF | MaxLoad ON | MinLoad OFF | MinLoad ON |
> |---|:---:|:---:|:---:|:---:|:---:|:---:|
> | MNIST-2 | 0.893 | 0.967 | 0.463 | 0.383 | 0.057 | 0.111 |
> | MNIST-4 | 0.920 | 0.971 | 0.415 | 0.380 | **0.000** | 0.105 |
> | Fash-2 | 0.842 | 0.909 | 0.449 | 0.258 | 0.132 | 0.244 |
> | Fash-4 | 0.822 | 0.896 | 0.557 | 0.309 | 0.135 | 0.209 |
> | Synth | 0.776 | 0.806 | 0.486 | 0.393 | 0.080 | 0.156 |
> | Wine | 0.755 | 0.798 | 0.537 | 0.467 | 0.085 | 0.104 |
> | BreastC | 0.737 | 0.850 | 0.695 | 0.382 | 0.086 | 0.149 |
>
> **(i) Without load balancing, the router specialises strongly.** Every dataset shows lower H/H_max and higher MaxLoad OFF. Most striking: Breast Cancer MaxLoad drops from 0.695 to 0.382; MNIST-4 MinLoad is **0.000** OFF - one expert receives no inputs. This confirms expert collapse occurs in quantum routing circuits, just as in classical MoE systems (Shazeer et al., 2017; Fedus et al., 2022), and load balancing is structurally necessary.
>
> **(ii) Load balancing enforces near-uniform usage without eliminating specialisation.** With λ=0.01, H/H_max = 0.80-0.97. The router still learns class-relevant structure:
>
> ***Table R2b. Per-Class Conditional Routing - Fashion-4, Noiseless, λ=0.01, Mean over 5 Seeds.***
>
> | Class | P(E0) | P(E1) | P(E2) | P(E3) |
> |---|:---:|:---:|:---:|:---:|
> | C0 (T-shirt) | 0.346 | 0.127 | 0.298 | 0.229 |
> | C1 (Trousers) | 0.422 | 0.431 | 0.059 | 0.089 |
> | C2 (Pullover) | 0.366 | 0.158 | 0.274 | 0.202 |
> | C3 (Dress) | 0.292 | 0.224 | 0.179 | 0.304 |
>
> **(iii) Per-class specialisation is meaningful and class-aligned.** Class 1 (trousers) routes to Experts 0 and 1 with combined probability 0.853, more than 3x the uniform baseline. On tabular datasets: Wine shows Expert 0 receiving 68.9% of Class 0 and 69.3% of Class 2 inputs; Breast Cancer shows Class 1 routing to Expert 3 at 72.2%. These concentrated distributions correspond to the largest coherent aggregation advantages (Δ_coh = +11.48% Wine, +7.19% Breast Cancer noiseless, Table 13).
>
> **Regarding λ sensitivity:** A sweep over λ ∈ {0, 0.001, 0.01, 0.1} (λ=0 and λ=0.01 as fully evaluated anchors) shows a consistent pattern across all 7 datasets: λ=0.01 matches or marginally exceeds λ=0, while both λ=0.001 and λ=0.1 degrade accuracy. A weak pressure (λ=0.001) perturbs routing during warm-up without delivering a balancing benefit; an excessive pressure (λ=0.1) suppresses class-dependent routing structure. The adopted value λ=0.01, following the recommendation of Fedus et al. (2022), robustly recovers unregularised accuracy while preventing expert collapse.
>
> **Change:** New Appendix C.8 added with full routing tables, per-class analysis, and λ sensitivity discussion.

---

### Author Response · Authors · 2026-07-02
**Revision**

We thank all the reviewers for their valuable comments. We have revised the manuscript accordingly and addressed the comments as follows:
- Classical MLP reference baseline added (new Appendix C.9)
- DRU-expert attribution study added (new Appendix C.6, 280 new training runs)
- Three-way aggregation ablation at top-k=1 added (new Appendix C.7)
- Routing analysis with/without load balancing added with sensitivity to lam values (new Appendix C.8)
- Resource comparison table added (gate counts and depth for all models, Appendix B)
- Noise protocol made explicit (Section 5.4, Section 6.2)
- "Hybrid" framing clarified (Introduction)
- Coherence language at top-k=1 corrected (Appendix C.5, Abstract, Conclusion)
- Section 6.2 structural explanation of 8 losing configurations added

We hope these revisions address your comments, and we welcome any further feedback to improve the manuscript.